# Low-latency automotive vision with event cameras

Daniel Gehrig[1✉] & Davide Scaramuzza[1✉]

The computer vision algorithms used currently in advanced driver assistance systems rely on image-based RGB cameras, leading to a critical bandwidth–latency trade-off for delivering safe driving experiences. To address this, event cameras have emerged as alternative vision sensors. Event cameras measure the changes in intensity asynchronously, offering high temporal resolution and sparsity, markedly reducing bandwidth and latency requirements[1]. Despite these advantages, event-camera-based algorithms are either highly efficient but lag behind image-based ones in terms of accuracy or sacrifice the sparsity and efficiency of events to achieve comparable results. To overcome this, here we propose a hybrid event- and frame-based object detector that preserves the advantages of each modality and thus does not suffer from this trade-off. Our method exploits the high temporal resolution and sparsity of events and the rich but low temporal resolution information in standard images to generate efficient, high-rate object detections, reducing perceptual and computational latency. We show that the use of a 20 frames per second (fps) RGB camera plus an event camera can achieve the same latency as a 5,000-fps camera with the bandwidth of a 45-fps camera without compromising accuracy. Our approach paves the way for efficient and robust perception in edge-case scenarios by uncovering the potential of event cameras[2].

Frame-based sensors such as RGB cameras face a bandwidth–latency trade-off: higher frame rates reduce perceptual latency but increase bandwidth demands, whereas lower frame rates save bandwidth at the cost of missing vital scene dynamics due to increased perceptual latency[3] (Fig. 1a). Perceptual latency measures the time between the onset of a visual stimulus and its readout on the sensor.

This trade-off is notable in automotive safety, in which reaction times are important. Advanced driver assistance systems record at 30–45 frames per second (fps) (refs. 4–9), leading to blind times of 22–33 ms. These blind times can be crucial in high-speed scenarios, such as detecting a fast-moving pedestrian or vehicle or a lost cargo. Moreover, when high uncertainties are present, for example, when traffic participants are partially occluded or poorly lit because of adverse weather conditions, these frame rates artificially prolong decision-making for up to 0.1–0.5 s (refs. 10–14). During this time, a suddenly appearing pedestrian (Fig. 1b) running at 12 kph would travel 0.3–1.7 m, whereas a car driving at 50 kph would travel 1.4–6.9 m.

Reducing this blind time is vital for safety. To address this, the industry is moving towards higher frame rate sensors, substantially increasing the data volume[5]. Current driverless cars collect up to 11 terabytes of data per hour, a number that is expected to rise to 40 terabytes (ref. 15). Although cloud computing offers some solutions, it introduces high network latency.

A promising alternative are event cameras, which capture per-pixel changes in intensity instead of fixed interval frames[1]. They offer low motion blur, a high dynamic range, spatio-temporal sparsity and a microsecond-level resolution with lower bandwidth and power usage[16,17]. They adapt to scene dynamics, providing low-latency and low-bandwidth advantages. However, the accuracy of event-based methods is currently limited by the inability of the sensors to capture slowly varying signals[18–20] and the inefficiency of processing methods that convert events to frame-like representations for analysis with convolutional neural networks (CNNs)[19,21–29]. This leads to redundant computation, higher power consumption and higher computational latency. Computational latency measures the time since a measurement was read out until producing an output.

We propose a new hybrid event- and frame-based object detector that combines a standard CNN for images and an efficient asynchronous graph neural network (GNN) for events (Fig. 2). The GNN processes events in a recursive fashion, which minimizes redundant computation and leverages key architectural innovations such as specialized convolutional layers, targeted skipping of events and a specialized directed event graph structure to enhance computational efficiency.

Our method leverages the advantages of event- and frame-based sensors, leveraging the rich context information in images and sparse and high-rate event information from events for efficient, high-rate object detections with reduced perceptual latency. In an automotive setting, it covers the blind time intervals of image-based sensors while keeping a low bandwidth. In doing so, it provides additional certifiable snapshots of reality that show objects before they become visible in the next image (Fig. 1c) or captures object movements that encode the intent or trajectory of traffic participants.

Our findings show that pairing a 20-fps RGB camera with an event camera can match the latency of a 5,000-fps camera but with the

[1]Robotics and Perception Group, University of Zurich, Zurich, Switzerland. ✉e-mail: dgehrig@ifi.uzh.ch; sdavide@ifi.uzh.ch

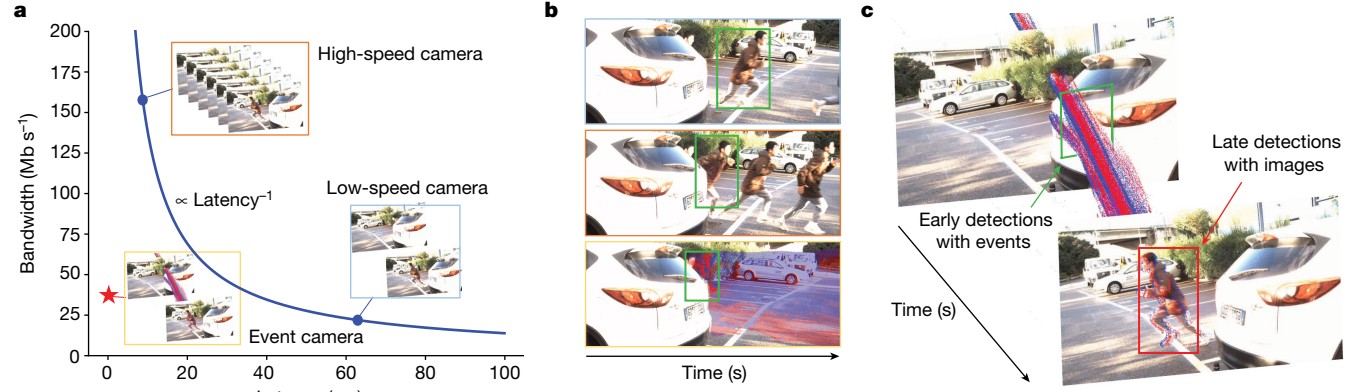

**Fig. 1 | Bandwidth–latency trade-off. a**, Unlike frame-based sensors, event cameras do not suffer from the bandwidth–latency trade-off: high-speed cameras (top left) capture low-latency but high-bandwidth data, whereas low-speed cameras (bottom right) capture low-bandwidth but high-latency data. Instead, our 20 fps camera plus event camera hybrid setup (bottom left, red and blue dots in the yellow rectangle indicate event camera measurements) can capture low-latency and low-bandwidth data. This is equivalent in latency to a 5,000-fps camera and in bandwidth to a 45-fps camera. **b**, Application scenario. We leverage this setup for low-latency, low-bandwidth traffic participant detection (bottom row, green rectangles are detections) that enhances the safety of downstream systems compared with standard cameras (top and middle rows). **c**, 3D visualization of detections. To do so, our method uses events (red and blue dots) in the blind time between images to detect objects (green rectangle), before they become visible in the next image (red rectangle).

bandwidth of a 45-fps camera, enhancing mean average precision (mAP) significantly (Fig. 4c). This approach harnesses the untapped potential of event cameras for efficient, accurate and fast object detection in edge-case scenarios.

## System overview

Our system, which we term deep asynchronous GNN (DAGr), is shown in Fig. 2. For a detailed visualization of each network component, see Extended Data Fig. 1, and for a visual explanation, see Supplementary Video 1. It combines a CNN[30], for image processing, with an asynchronous GNN[31,32], for processing of the events. These processing steps result in object detections with a high temporal resolution and a low latency (Fig. 2, green rectangles, bottom timeline).

We next discuss how events and images are combined. Each time an image arrives, the CNN processes it and shares the features with the asynchronous GNN in a unidirectional way, that is, the CNN features are shared with the GNN but not vice versa. The GNN thus leverages image features to boost its performance, especially when only a few events are triggered, as is common in static or slow-motion scenarios.

The asynchronous GNN constructs spatio-temporal graphs from events, following an efficient CUDA implementation inspired by ref. 32, and processes this graph together with features obtained from images (through skip connections) through a sequence of convolution and pooling layers. To facilitate both deep and efficient network training, we use graph residual layers[30] (Extended Data Fig. 1c). Moreover, we design a specialized voxel grid max pooling layer[33] (Extended Data Fig. 1d) that reduces the number of nodes in early layers and thus limits computation in lower layers. We mirror the detection head and training strategy of YOLOX[34], although we replace the standard convolution layers with graph convolution layers (Extended Data Fig. 1e). Finally, we design an efficient variant of the spline convolution layer[35] as a core building block. This layer pre-computes lookup tables and thus saves computation compared with the original layer in ref. 35.

To enhance efficiency, we follow the steps proposed in refs. 31,32,36 to convert the GNN to an asynchronous model. We first train the network on batches of events and images using the training strategy in ref. 34 and then convert the trained model into an asynchronous model by formulating recursive update rules. In particular, given an image $I_0$ and events $\mathcal{E}$ up to the next frame (50 ms later), we train the model to detect objects in the next frame.

The asynchronous model has the identical weights of the trained model but uses recursive update rules (Extended Data Fig. 2) to process events individually and produces an identical output. At each layer, it retains a memory of its previous graph structure and activation, which it updates for each new event. These updates are highly localized and thus reduce the overall computation by a large margin, as shown in refs. 31,32,36. To maximize the computation savings through this method, we adopt three main strategies. First, we limit the computation in each layer to single messages that are sent between nodes that had their feature or node position changed (Extended Data Fig. 2a), and these changes are then relayed to the next layer. Second, we prune non-informative updates, which stops the relaying of updates to lower layers (Extended Data Fig. 2b). This pruning step happens at max pooling operations, which are executed early in the network and thus maximize the potential of pruning. Finally, we use directed and undirected event graphs (Extended Data Fig. 2c). Directed event graphs connect only nodes if they are temporally ordered, which stifles update propagation and leads to further efficiency gains.

We report ablation studies on each component of our method in the Methods. Here we report comparisons of our system with state-of-the-art event- and frame-based object detectors both in terms of efficiency and accuracy. First, we show the performance of the asynchronous GNN when processing events alone before showing results with images and events. Then, we compare the ability of our method to detect objects in the blind time between consecutive frames. We find that our method balances achieving high performance—exceeding both image- and event-based detectors by using images—and remaining efficient, more so than existing methods that process events as dense frames.

## Using only events

We compare the GNN in our method with state-of-the-art dense and asynchronous event-based methods and report results in Fig. 3a,b. For a full table of results, see Extended Data Table 1. We enumerate the methods in the Methods. The metrics we report are the mAP, the average number of floating point operations (FLOPS) for each newly inserted event, and the average power consumption for computation.

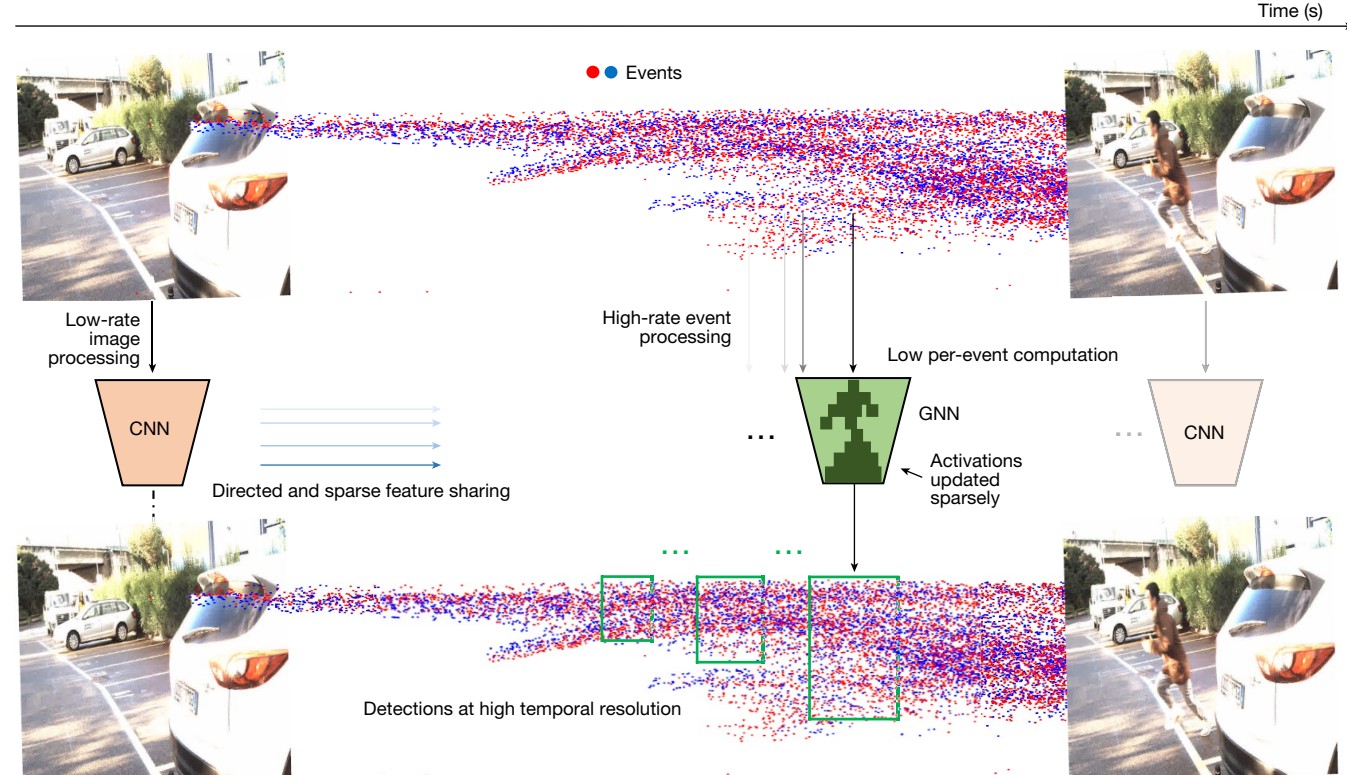

**Fig. 2 | Overview of the proposed method.** Our method processes dense images and asynchronous events (blue and red dots, top timeline) to produce high-rate object detections (green rectangles, bottom timeline). It shares features from a dense CNN running on low-rate images (blue arrows) to boost the performance of an asynchronous GNN running on events. The GNN processes each new event efficiently, reusing CNN features and sparsely updating GNN activations from previous steps.

To measure the power, we count the number of multiply accumulate operations (MACs) and multiply it by 1.69 pJ (ref. 37). We evaluate four versions of our model: nano (N), small (S), medium (M) and large (L). These differ in the number of features in the layer blocks 3, 4 and 5 and in the detection heads and have 32, 64, 92 and 128 channels in these layers, respectively.

According to Fig. 3, recurrent dense methods RED and ASTM-Net outperform our large model by 7.9 mAP and 14.6 mAP, respectively, but use more computation (4,712 compared with 1.36 for our nano-model). We believe that deeper networks and recurrence are two main factors that help performance in their methods. By contrast, our large model with 32.1 mAP outperforms the recurrent method MatrixLSTM (ref. 29) by 1.1 mAP and 120 times fewer FLOPS, and outperforms feedforward methods Events + RRC (ref. 38) (30.7 mAP), Inception + SSD (ref. 26) (30.1 mAP) and Events + YOLOv3 (ref. 27) (31.2 mAP). When compared with the spiking network Spiking DenseNet[39], we find that our method has a 13.1 point higher mAP. The low performance of the SNN is expected to increase as better learning strategies become available to the community. We find that our small-model outperforms all sparse methods in terms of computation, with around 13% times fewer million floating point operations (MFLOPS) per event than the runner-up AEGNN (ref. 31). It also achieves a 14.1-mAP higher performance than AEGNN. Our smallest network, nano, is 3.8 times more efficient while still outperforming AEGNN by 10 mAP. In terms of power consumption, our smallest model requires only 1.93 µJ per event, which is the lowest for all methods.

On the N-Caltech101 dataset, our small model outperforms the state-of-the-art dense and sparse methods, achieving 70.2 mAP, which is 5.9 mAP higher than the runner-up AsyNet (ref. 36) and uses less computation than the state-of-the-art AEGNN (ref. 31). Our large model achieves the highest score with 73.2 mAP. Our nano-model achieves the lowest computation of 2.28 MFLOPS per event, 3.25 times lower than AEGNN, with a 3.4% higher mAP.

## Using images and events

We evaluate the ability of our method to fuse images and events by validating its performance on our self-collected DSEC-Detection dataset. Details on the dataset and collection can be found in the Methods and ref. 40. Instructions on how to download the DSEC-Detection and visualize it can be found at https://github.com/uzh-rpg/dsec-det. We report the performance of our method and state-of-the-art event- and frame-based methods after seeing one image, and 50 ms of events after that image. We also report the computation in MFLOPS per inserted event in Fig. 3c. The results are computed over the DSEC-Detection test set. For a full table of results, including the power consumption per event in terms of µJ, see Extended Data Table 1.

We see that our baseline method with the ResNet-18 backbone reaches a 9.1 point higher mAP than the Inception + SSD (18.4 mAP) and Events + YOLOv3 (28.7 mAP) methods. We argue that this discrepancy comes from the better detection head as observed in ref. 34 and the suboptimal way of stacking events into event histograms[27]. Events + YOLOX outperforms our method, which is compared on the same ResNet-18 backbone (37.6 mAP for our method compared with 40.2 mAP for Events + YOLOX). This difference may come from the bidirectional feature sharing between event and frame features in Events + YOLOX, which is absent in our method. Finally, using a larger ResNet-50 backbone boosts our performance to 41.9 mAP. In terms of computational complexity, our method outperforms all methods, using only roughly 0.03% of the computation of the runner-up Events + YOLOX. The computational complexity is only weakly affected by the CNN backbone, decreasing as the capacity of the CNN backbone is increased.

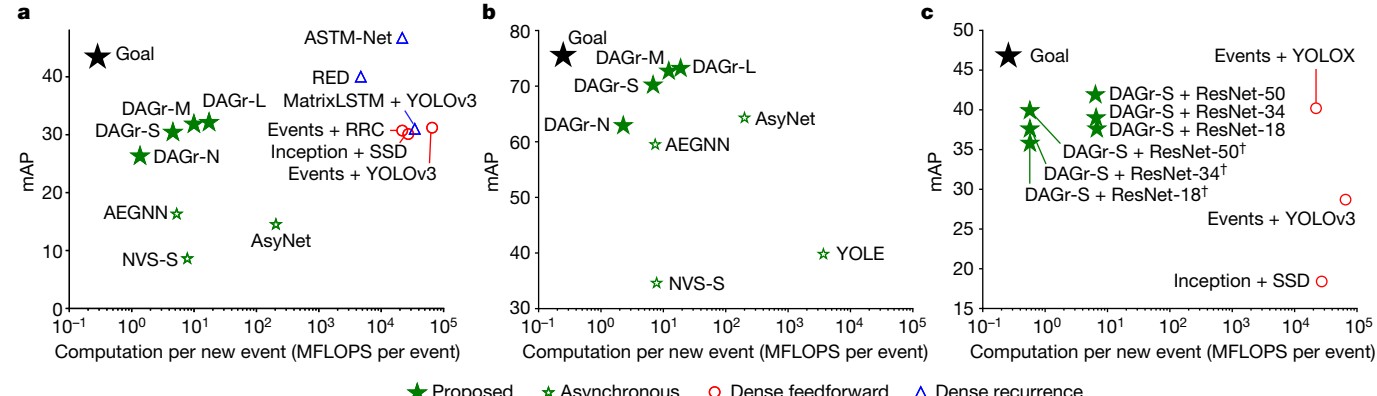

**Fig. 3 | Comparison summary of our method with state-of-the-art methods. a,b**, Comparison of asynchronous, dense feedforward and dense recurrent methods, in terms of task performance (mAP) and computational complexity (MFLOPS per inserted event) on the purely event-based Gen1 detection dataset[41]

(**a**) and N-Caltech101 (ref. 42) (**b**). **c**, Results of DSEC-Detection. All methods on this benchmark use images and events and are tasked to predict labels 50 ms after the first image, using events. Methods with dagger symbol use directed voxel grid pooling. For a full table of results, see Extended Data Table 1.

This indicates that event features are increasingly filtered as the image features become more important. Again, in terms of power consumption, our method outperforms all others by using only 5.42 μJ per event. Adopting directed edges (marked with dagger symbol) reduces the computation of our method with ResNet-50 backbone by 91% while incurring only an mAP reduction of 2%.

## Inter-frame detection performance

We report the detection performance of our method for different temporal offsets from the image $\frac{i}{n}\Delta t$ with $n = 10$ and $i = 0, \ldots, 10$ and $\Delta t = t_E - t_I = 50$ ms and evaluate on interpolated ground truth, described in the Methods. Here, $t_I$ denotes the frame time, and the start time of the event window inserted into the GNN, and $t_E$ denotes the end time of the event window. Note the ground truth here is limited to a subset for which no appearing or disappearing objects are present. We thus evaluate the ability of the method to measure both linear (between the interval) and nonlinear motions (at $t = 50$ ms), as well as complex object deformations. These arise especially in modelling pedestrians, which are frequently subject to sudden, complex and reflexive motion and have deformable appearances such as when

they stretch their arms, stumble or fall. We plot the detection performance for different temporal offsets in Fig. 4a, with and without events (cyan and yellow, respectively), and for the Events + YOLOX baseline (blue). For the image baseline, we also test with a constant and linear extrapolation model (yellow and brown). Whereas with the constant extrapolation model we keep object positions constant over time, for the linear model we perform a matching step with previous detections and then propagate the objects linearly into the future. More details on the linear extrapolation technique are given in the Methods. We also provide further results with different backbones in the Methods.

Our event- and image-based method (cyan) shows a slight performance increase throughout the 50-ms period, ending with a 0.7 mAP higher score after 50 ms. This is probably because of the addition of events, that is, more information becomes available. The subsequent slight decrease is probably because of the image information becoming more outdated. Events + YOLOX starts at a lower mAP of 34.7 before rising to 42.5 and settling at 42.2 at 50 ms. Notably, Events + YOLOX has an 8.8 mAP lower performance than our method at $t = 0$ and is less stable overall, gaining up to 7.5 mAP between 0 ms and 50 ms. Although all methods were trained with a fixed time window of 50 ms,

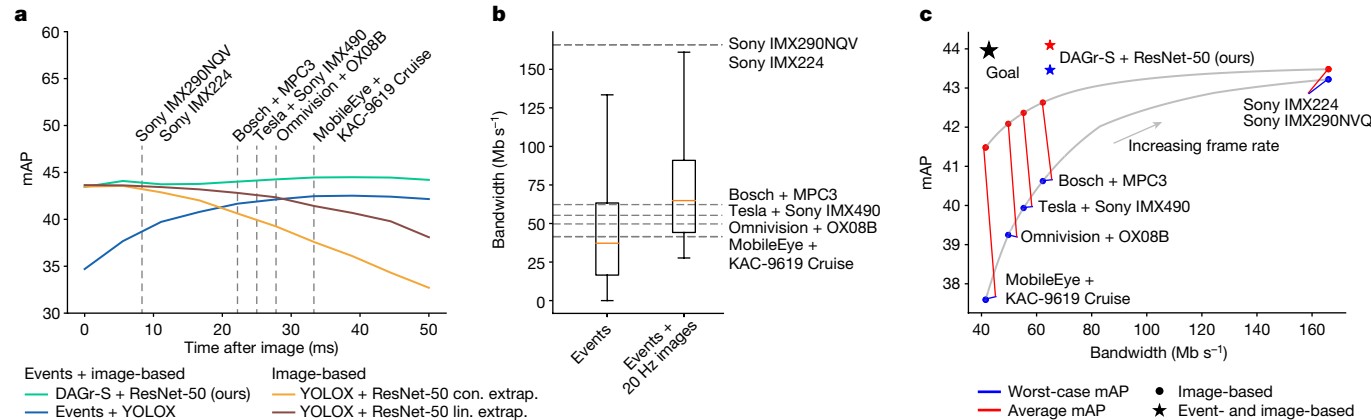

**Fig. 4 | Comparison of inter-frame detection performance for our method and state-of-the-art methods. a**, Detection performance in terms of mAP for our method (cyan), baseline method Events + YOLOX (ref. 34) (blue) and image-based method YOLOX (ref. 34) with constant and linear extrapolation (yellow and brown). Grey lines correspond to inter-frame intervals of automotive cameras. **b**, Bandwidth requirements of these cameras, and our hybrid event + image camera setup. The red lines correspond to the median, and the box

contains data between the first and third quartiles. The distance from the box edges to the whiskers measures 1.5 times the interquartile range. **c**, Bandwidth and performance comparison. For each frame rate (and resulting bandwidth), the worst-case (blue) and average (red) mAP is plotted. For frame-based methods, these lie on the grey line. The performance using the hybrid event + image camera setup is plotted as a red star (mean) and blue star (worst case). The black star points in the direction of the ideal performance–bandwidth trade-off.

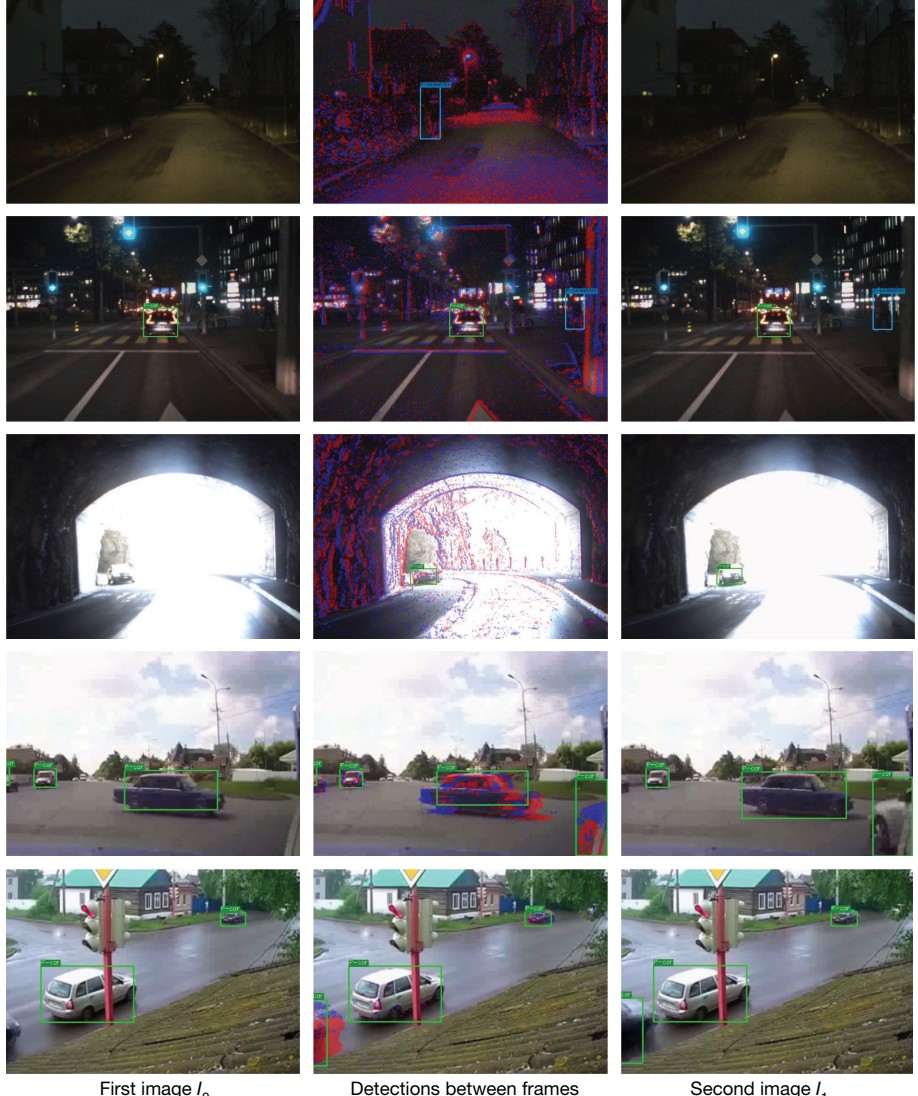

First image $I_0$        Detections between frames        Second image $I_1$

**Fig. 5 | Qualitative results of the proposed detector for edge-case scenarios.** The first column shows detections for the first image $I_0$. The second column shows detections between images $I_0$ and $I_1$ using events. The third column shows detections for the second image $I_1$. Detections of cars are shown by green rectangles, and of pedestrians by blue rectangles.

our method can more stably generalize to different time windows, whereas Events + YOLOX overfits to time windows close to 50 ms.

By contrast, the purely image-based method (yellow) with constant extrapolation degrades by 10.8 mAP after 50 ms. With linear extrapolation, this degradation is reduced to 6.4 mAP. This highlights the importance of using events as an extra information source to update predictions and provide certifiable snapshots of reality. Qualitative comparisons in Fig. 5 support the importance of events. In the first image (first column), some cars or pedestrians are not visible either because of image degradation (rows 1–3) or because they are outside the field of view (rows 4 and 5). Events from an event camera (second column) make objects that are poorly lit (rows 1–3) or just entering the field of view (rows 4 and 5) visible. In the second frame (third column), objects become visible in the next frame (rows 2–5); however, in scenarios such as in rows 4 and 5, the car has already undergone substantial movement, which may indicate a safety hazard in the immediate future. Using the events in Fig. 5 (second column) can provide valuable additional time to plan and increase safety. Moreover, we conclude from the results at time $t = 50$ ms that events improve object detection for nonlinearly moving or deformable objects over image-based approaches, even when considering linear extrapolation.

It is also crucial to put the increase in mAP into the context of the application. The mAP measures a weighted average of precisions at each detection threshold. At each threshold, the weight corresponds to the increase in recall from the previous threshold. The mAP is thus maximized if a method retains a high precision while the precision-recall curve undergoes a steep increase. We observe that using an event camera mostly aids in increasing the recall slope. The recall slope is increased because the addition of an event camera improves object localization between the frames. This improvement in localization entails a higher intersection over union and thus reduces false negatives at high thresholds. Reducing false negatives contributes to increased recall.

## Bandwidth–performance trade-off

The previous results show that low-frame-rate cameras yield lower mAP after the end of the frame interval. We characterize this mAP for different frame-rate sensors in Fig. 4a (grey lines). For the full list of compared automotive cameras, see Extended Data Table 2. Although a frame rate of 120 fps (ref. 6) leads to only a 0.7-mAP drop, 30 fps (refs. 4,9) leads to a 6.9-mAP drop. We plot this drop over the required bandwidth

(Fig. 4b) in Fig. 4c and also show our setup for comparison. The bandwidth was computed for an automotive-grade VGA RGB camera with a 12-bit depth (ref. 40) at different frame rates (see Extended Data Table 2 for a summary).

For each image sensor, we compute the minimum and average mAP within the range from $t = 0$ to $t = \Delta T$ seconds, where $\Delta T$ is the interframe interval corresponding with the worst-case and average performance. Note that the worst-case mAP indicates robustness in safety-critical situations. Our method outperforms YOLOX running with lower-frame-rate cameras in terms of accuracy while outperforming the high-frame-rate (120 fps) camera-based method in terms of both accuracy and bandwidth. Regarding worst-case and average mAP, our method outperforms YOLOX running on all different cameras. In particular, our method outperforms YOLOX running with the 45-fps MPC3 camera from Bosch[7] by 2.6 mAP, while requiring only 4% more data (64.9 Mb s$^{-1}$ compared with 62.3 Mb s$^{-1}$). It outperforms the 120 fps Sony IMX224 (ref. 6) by 0.2 mAP while requiring only 41% of the bandwidth. This finding shows that the combination of a 20-fps RGB camera with an event camera features a 0.2-ms perceptual latency, on par with that of a 5,000-fps RGB camera, but with only 4% more data bandwidth than a 45-fps automotive sensor (Fig. 4b). Figure 4a,c implies that this sensor combination does not incur a performance loss compared with high-speed standard cameras and can increase the worst-case performance by up to 2.6% over the 45-fps camera.

## Discussion

Leveraging the low latency and robustness of event cameras in the automotive sector requires a carefully designed algorithm that considers the different data structures of events and frames. We have presented DAGr, a highly efficient object detector that shows several advantages over state-of-the-art event- and image-based object detectors. First, it uses a highly efficient asynchronous GNN, which processes events as a streaming data structure instead of a dense one[26,27,34] and is thus four orders of magnitude more efficient. Second, it innovates on the architecture building blocks to scale the depth of the network while remaining more efficient than competing asynchronous methods[31,32]. As a result of a deeper network, our method can achieve higher accuracy compared with all other sparse methods. Finally, in combination with images, our method can effectively detect objects in the blind time between frames and maintain a high detection performance throughout this blind time, unlike competing baseline methods. Moreover, it can achieve this while remaining highly efficient, unlike other compared fusion methods that need to reprocess data several times[26,27,34] leading to wasteful computation.

Combining this approach with additional sensors such as LiDAR (light detection and ranging) sensors can present a promising future research direction. LiDARs, for example, can provide strong priors, which may increase the performance of our approach and reduce complexity if shallower networks are used.

Finally, although the current approach promises four orders of magnitude efficiency improvements over the state-of-the-art event- and image-based approaches, this does not yet translate to the same time efficiency gains. The current work improves the runtime performance of the algorithm by 3.7 over dense methods, but further runtime reductions must come from a suitable implementation on potentially spiking hardware accelerators.

Notwithstanding the remaining limitations and future work, demonstrating several orders of magnitude efficiency gains compared with traditional event- and image-based methods and leveraging images for robust and low-bandwidth, low-latency object detection represents a milestone in computer vision and machine intelligence. These results pave the way to efficient and accurate object detection in edge-case scenarios.

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

# Methods

In the first step, we will give a general overview of our hybrid neural network architecture, together with the processing model to generate high-rate object detections (see section 'Network overview'). Then we will provide more details about the asynchronous GNN (see section 'Deep asynchronous GNN') and will discuss the new network blocks that simultaneously push the performance and efficiency of our GNN. Finally, we will describe how our model is used in an asynchronous, event-based processing mode (see section 'Asynchronous operation').

## Network overview

An overview of the network is shown in Extended Data Fig. 1. Our method processes dense images and sparse events (red and blue dots, top left) with a hybrid neural network. A CNN branch $F_I$ processes each new image $I \in \mathbb{R}^{H \times W \times 3}$ at time $t_I$, producing detection outputs $\mathcal{D}^I$ and intermediate features $\mathcal{G}^I = \{g_l^I\}_{l=1}^L$ (blue arrows), where $l$ is the layer index. The GNN branch $F_E$ then takes image-based detection outputs, image features and event graphs constructed from raw events $\mathcal{E} = \{e_i | t_I < t_i < t_E\}$ with $t_I < t_E$ and events $e_i$, as input to generate detections for each time $t_E$. In summary, the detections at time $t_E$ are computed as

$$\mathcal{D}^I, \mathcal{G}^I = F_I(I) \tag{1}$$

$$\mathcal{D}^E = F_E(\mathcal{D}^I, \mathcal{G}^I, \mathcal{E}), \tag{2}$$

In normal operation, equation (1) is executed each time a new image arrives and essentially generates feature banks $\mathcal{D}^I$ and $\mathcal{G}^I$ that are then reused in equation (2). As will be seen later, $F_E$, being an asynchronous GNN, can be first trained on full event graphs, and then deployed to consume individual events in an incremental fashion, with low computational complexity and identical output to the batched form. As a result, the above equations describe a high-rate object detector that updates its detections for each new event. In the next section, we will have a closer look at our new GNN, before delving into the full hybrid architecture.

## Deep asynchronous GNN

Here we propose a new, highly efficient GNN, which we term, deep asynchronous GNN (DAGr). It processes events as spatio-temporal graphs. However, before we can describe it, we first give some preliminaries on how events are converted into graphs.

**Graph construction.** Event cameras have independent pixels that respond asynchronously to changes in logarithmic brightness **L**. Whenever the magnitude of this change exceeds the contrast threshold $C$, that pixel triggers an event $e_i = (\mathbf{x}_i, t_i, p_i)$ characterized by the position $\mathbf{x}_i$, timestamp $t_i$ with microsecond resolution and polarity (sign) $p_i \in \{-1, 1\}$ of the change. An event is triggered when

$$p_i[\mathbf{L}(\mathbf{x}_i, t_i) - \mathbf{L}(\mathbf{x}_i, t_i - \Delta t_i)] > C. \tag{3}$$

The event camera thus outputs a sparse stream of events $\mathcal{E} = \{e_i\}_{i=0}^{N-1}$. As in refs. 31,32,43–45, we interpret events as three-dimensional (3D) points, connected by spatio-temporal edges.

From these points, we construct the event graph $\mathcal{G} = \{\mathcal{V}, E\}$ consisting of nodes $\mathcal{V}$ and edges $E$. Each event $e_i$ corresponds to a node. These nodes $\mathbf{n}^i \in \mathcal{V}$ are characterized by their position $\mathbf{n}_p^i = (\hat{\mathbf{x}}_i, \beta t_i) \in \mathbb{R}^3$ and node features $\mathbf{n}_f^i = p_i \in \mathbb{R}$. Here $\hat{\mathbf{x}}_i$ is the event pixel coordinate, normalized by the height and width, and $t_i$ and $p_i$ are taken from the corresponding event. To map $t_i$ into the same range as $\mathbf{x}_i$, we rescale it by a factor of $\beta = 10^{-6}$. These nodes are connected by edges, $(i,j) \in E$, connecting nodes $\mathbf{n}_i$ and $\mathbf{n}_j$, each with edge attributes $e_{ij} \in \mathbb{R}^{d_e}$. We connect nodes that are temporally ordered and within a spatio-temporal distance from each other:

$$(i,j) \in E \quad \text{if} \quad \|\mathbf{n}_p^i - \mathbf{n}_p^j\|_\infty < R \quad \text{and} \quad t_i < t_j. \tag{4}$$

Here $\| \cdot \|_\infty$ returns the absolute value of the largest component. For each edge, we associate edge features $e_{ij} = (\mathbf{n}_{xy}^j - \mathbf{n}_{xy}^i)/2r + 1/2$. Here, $\mathbf{n}_{xy}$ denote the $x$ and $y$ components of each node, and $r$ is a constant, such that $e_{ij} \in [0, 1]^2$. Constructing the graph in this way gives us several advantages. First, we can leverage the queue-based graph construction method in ref. [32] to implement a highly parallel graph construction algorithm on GPU. Our implementation constructs full event graphs with 50,000 nodes in 1.75 ms and inserts single events in 0.3 ms on a Quadro RTX 4000 laptop GPU. Second, the temporal ordering constraint above makes the event graph directed[32,45], which will enable high efficiency in early layers before pooling (see section 'Asynchronous operation'). In this work, we select $R = 0.01$ and limit the number of neighbours of each node to 16.

**Deep asynchronous GNN.** In this section, we describe the function $F_E$ in equation (2). For simplicity, we first describe it without the fusion terms $\mathcal{D}^I$ and $\mathcal{G}^I$ and describe only how processing is performed on events alone. We later give a complete description, incorporating fusion.

An overview of our neural network architecture is shown in Extended Data Fig. 1. It processes the spatio-temporal graphs from the previous section and outputs object detection at multiple scales (top right). It consists of five alternating residual layers (Extended Data Fig. 1c) and max pooling blocks (Extended Data Fig. 1d), followed by a YOLOX-inspired detection head at two scales (Extended Data Fig. 1e). Crucially, our network has a total of 13 convolution layers. By contrast, the methods in ref. 32 and ref. 31 feature only five and seven layers, respectively, making our network almost twice as deep as the previous methods. Before each residual layer, we concatenate the $x$ and $y$ coordinates of the node position onto the node feature, which is indicated by +2 at the residual layer input. Residual layers and the detection head use the lookup table-based spline convolutions (LUT-SCs) as the basic building block (Extended Data Fig. 1f). These LUT-SCs are trained as a standard spline convolution[31,35] and later deployed as an efficient lookup table (see section 'Asynchronous operation').

*Spline convolutions.* Spline convolutions, shown in Extended Data Fig. 1f, update the node features by aggregating messages from neighbouring nodes:

$$\mathbf{n}_f'^i = W\mathbf{n}_f^i + \sum_{(j,i) \in E} W(e_{ij})\mathbf{n}_f^j, \quad \text{and} \quad \mathbf{n}_p'^i = \mathbf{n}_p^i. \tag{5}$$

Here $\mathbf{n}_f'^i$ is the updated feature at node $\mathbf{n}_i$, $W \in \mathbb{R}^{c_{out} \times c_{in}}$ is a matrix that maps the current node feature $\mathbf{n}_f^i$ to the output, and $W(e_{ij}) \in \mathbb{R}^{c_{out} \times c_{in}}$ is a matrix that maps neighbouring node features $\mathbf{n}_f^j$ to the output. In ref. 35, $W(e_{ij})$ is a matrix-valued smooth function of the edge feature $e_{ij}$. Remember that the edge features $e_{ij} \in [0, 1]^2$, which is the domain of $W(e_{ij})$. The function $W(e_{ij})$ is modelled by a $d$-order B-spline in $m = 2$ dimensions with $k \times k$ learnable weight matrices equally spaced in $[0, 1]^2$. During the evaluation, the function interpolates between these learnable weights according to the value of $e_{ij}$. In this work, we choose $d = 1$ and $k = 5$.

*Max pooling.* Max pooling, shown in Extended Data Fig. 1d, splits the input space into $g_x \times g_y \times g_t$ voxels $V$ and clusters nodes in the same voxel. At the output, each non-empty voxel has a node, located at the rounded mean of the input node positions and with its feature equal to the maximum of the input nodes features.

$$\mathbf{n}_f'^i = \max_{\mathbf{n} \in V_i} \mathbf{n}_f, \quad \text{and} \quad \mathbf{n}_p'^i = \frac{1}{\alpha}\left[\frac{\alpha}{|V_i|} \sum_{\mathbf{n} \in V_i} \mathbf{n}_p\right]. \tag{6}$$

Here multiplying by $\alpha = \left[H, W, \frac{1}{\beta}\right]^\top$ scales the mean to the original resolution. To compute the new edges, it forms a union of all edges

connecting the cluster centres and removes duplicates. Formally, the edge set of the output graph after pooling, $E'_{pool}$, is computed as

$$E'_{pool} = \{e_{c_i c_j} | e_{ij} \in E\}. \tag{7}$$

Here $c_i$ retrieves the index of the voxel in which the node $\mathbf{n}^i$ resides, and duplicates are removed from the set. This operation can result in bidirectional edges between output nodes if at least one node from voxel A is connected to one of voxel B and vice versa. The combination of max pooling and position rounding has two main benefits: first, it allows the implementation of highly efficient LUT-SC, and second, it enables update pruning, which further reduces computation, discussed in the section 'Events only' under 'Ablations'. For our pooling layers, we select $(g_x, g_y, g_t)_i = (56/2^i, 40/2^i, 1)$, where $i$ is the index of the pooling layer. As seen in this section, selecting $g_t = 1$ is crucial to obtain high performance because it accelerates the information mixing in the network.

*Directed voxel grid pooling.* As previously mentioned, the constructed event graph has a temporal ordering, which means that the edges pass only from older to newer nodes. Although this property is conserved in the first few layers of the GNN, after pooling it is lost to a certain extent. This is because edge pooling, described in equation (7), has the potential to generate bidirectional edges (Extended Data Fig. 2d, top). Bidirectional edges are formed when there is at least one edge going from voxel A to voxel B, and one edge going from voxel B to voxel A, such that pooling merges them into one bidirectional edge between A and B. Although bidirectional edges facilitate the distribution of messages throughout the network and thus boost accuracy, they also increase computation during asynchronous operation significantly. This is because bidirectional edges grow the k-hop subgraph that needs to be recomputed at each layer. In this work, we introduce a specialized directed voxel pooling, which instead curbs this growth, by eliminating bidirectional edges from the output, thus creating temporally ordered graphs at all layers. It does this, by redefining the pooling operations. Although feature pooling is the same, position pooling becomes

$$\mathbf{n}'^i_t = \max_{\mathbf{n} \in V_i} \mathbf{n}_t \quad \text{and} \quad \mathbf{n}'^i_{xy} = \frac{1}{\alpha}\left[\frac{\alpha}{|V_i|}\sum_{\mathbf{n} \in V_i} \mathbf{n}_{xy}\right]. \tag{8}$$

Here we pool the coordinates $x$ and $y$ using mean pooling and timestamps $t$ with max pooling. We then redefine the edge pooling operation as

$$E'_{dpool} = \{e_{c_i c_j} | e_{ij} \in E \quad \text{and} \quad \mathbf{n}^{c_j}_t > \mathbf{n}^{c_i}_t\}, \tag{9}$$

where we now impose that edges between output nodes can exist only if the timestamp of the source node is smaller than that of the destination node. This condition essentially acts as a filter on the total number of pooled edges. As will be discussed later, this pooling layer increases the efficiency, while also affecting the performance. However, we show that when combined with images (see section 'Images and events'), this pooling layer can manifest both high accuracy and efficiency.

*Detection head.* Inspired by the YOLOX detection head, we design a series of (LUT-SC, BN and ReLU) blocks that progressively compute a bounding box regression $\mathbf{f}_{reg} \in \mathbb{R}^4$, class score $\mathbf{f}_{cls} \in \mathbb{R}^{n_{cls}}$ and object score $\mathbf{f}_{obj} \in \mathbb{R}$ for each output node. We then decode the bounding box location as in ref. 34 but relative to the voxel location in which the node resides. This results in a sparse set of output detections.

Now that all components of the GNN are discussed, we will introduce the fusion strategy that combines the CNN and GNN outputs.

## CNN branch and fusion

The CNN branch $F_I$ (Extended Data Fig. 1) is implemented as a classical CNN, here ResNet[30], pretrained on ImageNet[46], whereas the GNN has the architecture from the section 'Deep asynchronous GNN'.

To generate the image features $\mathcal{G}^I$ used by the GNN, we process the features after each ResBlock with a depthwise convolution. To generate the detection output, we also apply a depthwise convolution to the last two scales of the output before using a standard YOLOX detection head[34]. We fuse features from the CNN with those from the GNN with sparse directed feature sampling and detection adding.

**Feature sampling.** Our GNN makes use of the intermediate image feature maps $\mathcal{G}^I$ using a feature sampling layer (Extended Data Fig. 1b), which, for each graph node, samples the image feature at that layer at the corresponding node position and concatenates it with the node feature. In summary, at each GNN layer, we update the node features with features derived from $\mathcal{G}^I$ by taking into account the spatial location of nodes in the image plane:

$$\hat{g}^i_l = g^I_l(\mathbf{n}^i_p) \tag{10}$$

$$\hat{\mathbf{n}}^i_f = \left[\hat{g}^i_l \| \mathbf{n}^i_f\right], \tag{11}$$

where $\hat{\mathbf{n}}^i_f$ is the updated node feature of node $\mathbf{n}^i$. Equation (10) samples image features at each event node location and equation (11) concatenates these features with the existing node features. Note that equations (10) and (11) can be done in an event-by-event fashion.

**Detection adding.** Finally, we add the outputs of the corresponding detection heads of the two branches. We do this before the decoding step[34], which applies an exponential map to the regression outputs and sigmoid to the objectness scores. As the outputs of the GNN-based and CNN-based detection heads are sparse and dense, respectively, care must be taken when adding them together. We thus initialize the detections at $t_E$ with $\mathcal{D}^I$ and then add the detection outputs of the GNN to the pixels corresponding to the graph nodes. This operation is also compatible with event-by-event updating of the GNN-based detections.

Detection adding is an essential step to overcome the limitations of event-based object detection in static conditions, because then the RGB-based detector can provide an initial guess even when no events are present. It also guarantees that in the absence of events, the performance of the method is lower bounded by the performance of the image-based detector.

**Training procedure.** Our hybrid method consists of two coupled object detectors that generate detection outputs at two different timestamps: one at the timestamp of the image $t_I$ and the other after observing events until time $t_E$ (Extended Data Fig. 1). As our labels are collocated with the image frames, this enables us to define a loss in both instances. We found that the following training strategy produced the best results: pretraining the image branch with the image labels first, then freezing the weights and training the depthwise convolutions and DAGr branch separately on the event labels.

As both branches are trained to predict detections separately, the DAGr network essentially learns to update the detections made by the image branch. This means that DAGr learns to track, detect and forget objects from the previous view.

## Asynchronous operation

As in refs. 31,32,36, after training, we deploy our hybrid neural network in an asynchronous mode, in which instead of feeding full event graphs, we input only individual events. Local recursive update rules are formulated at each layer that enforces that the output of the network for each new event is identical to that of the augmented graph that includes the old graph and the new event. As seen in refs. 31,32,36, the rules update only a fraction of the activations at each layer, leading to a drastic reduction in computation compared with a dense forward pass.

In this section, we will describe the steps that are taken after training to perform asynchronous processing.

**Initialization.** The conversion to asynchronous mode happens in three steps: (1) precomputing the image features; (2) LUT-SC caching and batch norm fusing; and (3) network activation initialization.

As a first step, when we get an image, we precompute the image features by running a forward pass through the CNN and applying the depthwise convolutions. This results in the image feature banks $\mathcal{G}^I$ and detections $\mathcal{D}^I$.

In the second step (LUT-SC caching), spline convolutions generate the highest computational burden in our method because they involve evaluating a multivariate, matrix-valued function and performing a matrix–vector multiplication. Following the implementation in ref. 35, computing a single message between neighbours requires

$$C_{msg} = (2[d+1]^m - 1)c_{in}c_{out} + (2c_{in} - 1)c_{out}, \qquad (12)$$

floating point operations (FLOPS), in which the first term computes the interpolation of the weight matrix and the second computes the matrix–vector product. Here the first term dominates because of the highly superlinear dependence on $d$ and $m$. Our LUT-SC eliminates this term. We recognize that the edge attributes $e_{ij}$ depend only on the relative spatial node positions. As events are triggered on a grid, and the distance between neighbours is bounded, these edge attributes can only take on a finite number of possible values. Therefore, instead of recomputing the interpolated weight at each step, we can precompute all weight matrices once and store them in a lookup table. This table stores the relative offsets of nodes together with their weight matrix. We thus replace the message propagation equation with

$$\mathbf{n}_f'^i = W\mathbf{n}_f^i + \sum_{(j,i)\in E} W_{ij}\mathbf{n}_f^j \qquad (13)$$

$$W_{ij} = \mathrm{LUT}(dx, dy), \qquad (14)$$

where $dx$ and $dy$ are the relative two-dimensional (2D) positions of nodes $i$ and $j$. Note that this transformation reduces the complexity of our convolution operation to $C_{msg} = (2c_{in} - 1)c_{out}$, which is on the level of the classical graph convolution (GC) used in ref. 32. However, crucially, LUT-SC still retains the relative spatial awareness of spline convolutions, as $W_{ij}$ change with the relative position and is thus more expressive than GCs. After caching, we fuse the weights computed above with the batch norm layer immediately following each convolution, thereby eliminating its computation from the tally. After pooling, ordinarily, node positions would not have the property that they lie on a grid anymore, as their coordinates get set to the centroid location. However, because we apply position rounding, we can apply LUT-SC caching in all layers of the network.

In the third step (network activation initialization), before asynchronous processing, we pass a dense graph through our network and cache the intermediate activations at each layer. Although in convolution layers we cache the activation, that is, the results of sums computed from equation (13), in max pooling layers we cache (1) the indices of input nodes used to compute the output feature for each voxel; (2) a list of currently occupied output voxels; and (3) a partial sum of node positions and node counts per voxel to efficiently update output node positions after pooling.

**Update propagation.** When a new event is inserted, we compute updates to all relevant nodes in all layers of the network. The goal of these updates is to achieve an output identical to the output the network would have computed if the complete graph with one event added was processed from scratch. The updates include (1) adding and recomputing messages in equation (13) if a node position or feature has

changed; (2) recomputing the maximum and node position for output nodes after each max pooling layer; and (3) adding and flipping edges when new edges are formed at the input. We will discuss these updates in the following sections. To facilitate computation, at each layer, we maintain a running list of unchanged nodes (grey) and changed nodes (cyan) and whether their position has changed, the feature has changed or both. The propagation rules are outlined in Extended Data Fig. 2.

*Convolution layers.* In a convolution layer (Extended Data Fig. 2a), if the node has a different position (Extended Data Fig. 2a, top), we recompute that feature of the node and resend a message from that node to all its neighbours. These are marked as green and orange arrows in Extended Data Fig. 2a (top row). If instead, only the feature of the node changed (Extended Data Fig. 2a, bottom), we update only the messages sent from that node to its neighbours. We can gain an intuition for these rules from equation (13). A change in the node feature $\mathbf{n}_f$ changes only one term in the sum that has to be recomputed. Instead, a node position change causes all weight matrices $W_{ij}$ to change, resulting in a recomputation of the entire sum.

*Pooling layers.* Pooling layers update only output nodes for which at least one input node has a changed feature or changed position. For these output nodes, the position and feature are recomputed using equation (6). Special care must be taken when using directed voxel pooling layers. Sometimes it can happen that an edge at the output of this layer needs to be inverted such that temporal ordering is conserved. In this case, the next convolution layer must compute two messages (Extended Data Fig. 2e), one to undo the first message and the other corresponding to the new edge direction. In this case, two nodes are changed instead of only one. However, edge inversion happens rarely and thus does not contribute markedly to computation.

**Reducing computation.** In this section, we describe various considerations and algorithms for reducing the computation of the two basic layers described above.

*Directed event graph.* As previously discussed, using a directed event graph notably reduces computation, as it reduces the number of nodes that need to be updated at each layer. We illustrate this concept in Extended Data Fig. 2c, in which we compare update propagation in graphs that are directed or possess bidirectional edges. Note that we encounter directed graphs either at the input layer (before the first pooling) or after directed voxel pooling layers. Instead, graphs with bidirectional edges are encountered after regular voxel pooling layers. As seen in Extended Data Fig. 2c (top), directed graphs keep the number of messages that need to be updated in each layer constant, as no additional nodes are updated at any layer. Instead, bidirectional edges send new messages to previously untouched nodes, leading to a proliferation of update messages, and as a result, computation.

*Update pruning.* Even when input nodes to a voxel pooling layer change, the output position and feature may stay the same, even after recomputation. If this is the case, we simply terminate propagation at that node, called update pruning, and thus save significantly in terms of computation. We show this phenomenon in Extended Data Fig. 2b. This can happen when (1) the rounding operation in equation (6) simply rounds a slightly updated position to the same position as before; and (2) the maximal features at the output belong to input nodes that have not been updated. Let us state the second condition more formally. Let $\mathbf{n}_{f,j}'^i$ be the $j$th entry of the feature vector belonging to the $i$th output node. Now let

$$\mathbf{n}^{k_j^i} = \arg\max_{\mathbf{n}\in V_i} \mathbf{n}_{f,j} \qquad (15)$$

be the input node for which the feature $\mathbf{n}_{f,j}$ at the $j$th position is maximal. The index $k_j^i$ selects this node from the voxel $V_i$. Thus, we may rewrite the equation for max pooling for each component as

$$\mathbf{n}_{f,j}'^i = \mathbf{n}_{f,j}^{k_j^i}. \qquad (16)$$

This means that essentially, only a subset of input nodes in the voxel contributes to the output, and this subset is exactly

$$\mathcal{P}_i = \{\mathbf{n}^{k_j^i} | j = 0, \ldots, c-1\} \subset V_i. \tag{17}$$

Moreover, as these nodes are indexed by $j$, and the $k_j^i$ could repeat, we know that the size of this subset satisfies $|\mathcal{P}_i| \leq c$, where $c$ is the number of features. We thus find that output features do not change if none of the changed inputs nodes to a given output node are within the set $\mathcal{P}_i$.

Thus, for each output node, we check the following conditions to see if update pruning can be performed. For all input nodes that have a changed position or feature, we check if (1) the changed node is currently in the set of unused nodes (greyed out in Extended Data Fig. 2b); (2) the changed feature of the node does not beat the current maximum at any feature index; and (3) its position change did not deflect the average output node position sufficiently to change rounding. If not all three conditions are met, we recompute the output feature for that node, otherwise, we prune the update and skip the computation in the lower layers. Skipping happens surprisingly often. In our case, we found that 73% of updates are skipped because of this mechanism. This also motivated us to place the max pooling layer in the early layers, as it has the highest potential to save computation. In a later section, we will show the impact these features have on the computation of the method.

*Simplification of concatenation operation.* During feature fusion in the hybrid network, owing to the concatenation of node-level features with image features (equation (11)), the number of intermediate features at the input to each layer of the GNN increases. This would essentially increase the computation of these layers. However, we apply a simplification, which significantly reduces this additional cost. Note that from equation (13) the output of the layer after the concatenation becomes

$$\begin{aligned}
\mathbf{n}_f'^i &= W\widehat{\mathbf{n}}_f^i + \sum_{(j,i)\in E} W_{ij}\widehat{\mathbf{n}}_f^j \\
&= W[\widehat{g}_l^i \| \mathbf{n}_f^i] + \sum_{(j,i)\in E} W_{ij}[\widehat{g}_l^j \| \mathbf{n}_f^j] \\
&= W^g\widehat{g}_l^i + W^f\mathbf{n}_f^i + \sum_{(j,i)\in E} W_{ij}^g\widehat{g}_l^j + \sum_{(j,i)\in E} W_{ij}^f\mathbf{n}_f^j \\
&= \underbrace{W^g\widehat{g}_l^i + \sum_{(j,i)\in E} W_{ij}^g\widehat{g}_l^j}_{\text{affected by } \mathbf{n}_p \text{ change}} + \underbrace{W^f\mathbf{n}_f^i + \sum_{(j,i)\in E} W_{ij}^f\mathbf{n}_f^j}_{\text{affected by } \mathbf{n}_p \text{ and } \mathbf{n}_f \text{ change}}.
\end{aligned} \tag{18}$$

In the equation above, we made use of the fact that weight matrix $W_{ij} = [W_{ij}^g \| W_{ij}^f]$ and thus multiplication results in the sum of products $W_{ij}^g\widehat{g}_l^i$ and $W_{ij}^f\mathbf{n}_f$. Note that this simplification does not imply that a similar operation could be performed with a pure depthwise convolution and addition of features, as the weight matrices $W_{ij}$ change for each neighbour. During asynchronous operation, the terms on the left need to be recomputed when there is a node position change, and the terms on the right need to be recomputed when there is a node position or node feature change. At most one node experiences a node position change in each layer, and thus the terms on the left do not need to be recomputed often.

## Datasets

**Purely event-based datasets.** We evaluate our method on the N-Caltech101 detection[42], and the Gen1 Detection Dataset[41]. N-Caltech101 consists of recording by a DAVIS240 (ref. 17) undergoing a saccadic motion in front of a projector, projecting samples of Caltech101 (ref. 47) on a wall. In post-processing, bounding boxes around the visible boxes were hand placed. The Gen1 Detection Dataset is a more challenging, large-scale dataset targeting an automotive setting. It was recorded with an ATIS sensor[48] with a resolution of $304 \times 240$, two classes, 228,123 annotated cars and 27,658 annotated pedestrians. As in ref. 19, we remove bounding boxes with diagonals below 30 and sides below 20 pixels from Gen1.

**Event- and image-based dataset.** We curate a multimodal dataset for object detection by using the DSEC[40] dataset, which we term DSEC-Detection. A preview of the dataset can be seen in Extended Data Fig. 6a.

It features data collected from a stereo pair of Prophesee Gen3 event cameras and FLIR Blackfly S global shutter RGB cameras recording at 20 fps. We select the left event camera and left RGB camera and align the RGB images with the distorted event camera frame by infinite depth alignment. Essentially, we first undistort the camera image, then rotate it into the same orientation as the event camera and then distort the image. The resulting image features only a maximal disparity of roughly 6 pixels for close objects at the edges of the image plane owing to the small baseline (4.5 cm). As object detection is not a precise per-pixel task, this kind of misalignment is sufficient for sensor fusion.

To create labels, we use the QDTrack[49,50] multiobject tracker to annotate the RGB images, followed by a manual inspection and removal of false detections and tracks. Using this method, we annotate the official training and test sets of DSEC[40]. Moreover, we label several sequences for the validation set and one complex sequence with pedestrians for the test set. We do this because the original dataset split was chosen to minimize the number of moving objects. However, this excludes cluttered scenes with pedestrians and moving cars. By including these additional sequences, we thus also address more complex and dynamic scenes. A detailed breakdown and comparison of the number of classes, instances per class and the number of samples are given in Extended Data Fig. 6b. Our dataset is the only one to feature images and events and consider semantic classes, to the best of our knowledge. By contrast, refs. 19,41 have only events, and ref. 51 considers only moving objects, that is, does not provide class information, or omits stationary objects.

**Statistics of edge cases.** We compute the percentage of edge cases for the DSEC-Detection dataset. We will define an edge case as an image that contains at least one appearing or disappearing object, which presumably would be missed by using a purely image-based algorithm. We found that this proportion is 31% of the training set and 30% of the test set. Moreover, we counted the number of objects that suddenly appear or disappear. We found that in the training set, 4.2% of objects disappear and 4.2% appear, whereas in the test set, 3.5% appear and 3.5% disappear.

**Comments on time synchronization.** Events and frames were hardware synchronized by an external computer that sent trigger signals simultaneously to the image and event sensor. While the image sensor would capture an image with a fixed exposure on triggering, the event camera would record a special event that exactly marked the time of triggering. We assign the timestamp of this event (and half an exposure time) to the image. We found that this synchronization accuracy was of the order of 78 µs, which we determined by measuring the mean squared deviation of the frame timestamps from a nominal 50,000 µs. More details can be found in ref. 40.

**Comments on network and event transport latencies.** As discussed earlier, we estimate the mean synchronization error of the order of 78 µs with hardware synchronization. Moreover, in a real-time system, the event camera will experience event transport delays that are split into a maximal sensor latency, MIPI to USB transfer latency and a USB to computer transfer latency, as discussed in ref. 52. For the Gen3 sensor, the sum of all worst-case latencies can be as low as 6 ms. It can be further reduced by using directly an MIPI interface in which case this latency reduces to 4 ms. However, this worst-case delay is achieved only during static scenarios, in which there is an exceptionally low event rate such that MIPI packets are not filled sufficiently. However, this case is rarely achieved because of the presence of sensor noise and also does

not affect dynamic scenarios with high event rates. More details can be found in ref. 53. Finally, note that although all three latencies would affect a closed-loop system, our work is evaluated in an open loop and thus does not experience these latencies, or synchronization errors due to these latencies.

In view of integrating our method into a multi-sensor system, which uses the network-based time synchronization standard IEEE1588v2, we analyse how the method performs when small synchronization errors between images and events are present. To test this, we introduce a fixed time delay $\Delta t_d \in [-20, 20]$ ms between the event and image stream. Note that for a given stimulus a delay of $\Delta t_d < 0$ denotes that events arrive earlier than images, whereas $\Delta t_d > 0$ denotes that events arrive later than images. We report the performance of DAGr-S + ResNet-50 on the DSEC-Detection test set in Extended Data Fig. 3b. As can be seen, our method is robust to synchronization errors up to 20 ms, suffering only a maximal performance decrease of 0.5 mAP. Making our method more robust to such errors remains the topic of further work.

**Comment on event-to-image alignment.** Throughout the dataset, event-to-image misalignment is small and never exceeds 6 pixels, and this is further supported by visual inspection of Extended Data Fig. 6a. Nonetheless, we characterize the accuracy that a hypothetical decision-making system would have if worst-case errors were considered. Consider a decision-making system that relies on accurate and low-latency positioning of actors such as cars and pedestrians. This system could use the proposed object detector (using the small-baseline stereo setup with an event and image camera) as well as a state-of-the-art event camera-based stereo depth method[54] (using the wide-baseline stereo event camera setup) to map a conservative region around a proposed detection. This system would still have a low latency and provide a low depth uncertainty because of a low disparity error of 1.2–1.3 pixels, characterized on DSEC in ref. 40.

We can calculate the depth uncertainty due to the stereo system as $\sigma_D = \frac{D^2}{fb_w}\sigma_d$. With a maximal disparity uncertainty $\sigma_d = 1.3$ pixels, the depth $D$ at 3 m, the focal length at $f = 581$ pixels and the event camera to event camera baseline at $b_w = 50$ cm. This results in a depth uncertainty of $\sigma_D = 4$ cm. Likewise, the lateral positioning uncertainty (due to shifted events) is $\sigma_l = \frac{D}{f}\sigma_d$.

For lateral positioning, we can assume a disparity error that is bounded by the misalignment between events and frames, which is $\sigma_d < \frac{fb_s}{D}$ where $b_s = 4.5$ cm is the small baseline between the event and image camera. Inserting this uncertainty, the resulting lateral uncertainty is bounded by $\sigma_p = \frac{D}{f}\sigma_d < \frac{D}{f}\frac{fb_s}{D} = b_s$, which means $\sigma_p < 4.5$ cm. These numbers are well within the tolerance limits of automotive systems that typically expect a 3% of distance to target uncertainty, which for 3 m would be 9 cm. Moreover, this lies within the tolerance limit of the current agent-forecasting methods[10–12] that are currently finding their way into commercial patents[13], in which we see displacement errors in prediction of the order of 0.6 m, more than one order of magnitude higher than the worst-case error of our system.

Finally, we argue that despite the misalignment, our object detector learns to implicitly realign events to the image frame because of the training setup. As the network is trained with object detection labels that are aligned with the image frame, and slightly misaligned events, the network learns to implicitly realign the events to compensate for the misalignment. As the misalignment is small, this is simple to learn. To test this hypothesis, we used the LiDAR scans in DSEC to align the object detection labels with the event stream, that is, in the frame it was not trained for, and observed a performance drop from 41.87 mAP to 41.8 mAP. First, the slight performance drop indicates that we are moving the detection labels slightly out of distribution, thus confirming that the network learns to implicitly apply a correction alignment. Second, the small magnitude of the change highlights that the misalignment is small.

**Ground truth generation for inter-frame detection.** To evaluate our method between consecutive frames, we generate ground truth as follows. We generate ground truth for multiple temporal offsets $\frac{i}{n}\Delta t$ with $n = 10$ and $i = 0, ..., 10$ and $\Delta t = t_E - t_I = 50$ ms. We then remove the samples from our dataset in which two consecutive images do not share the same object tracks and generate inter-frame labels by linearly interpolating the position ($x$ and $y$ coordinates of the top left bounding box corner) and size (height and width) of each object. We then aggregate detection evaluations at the same temporal offset across the dataset.

**Comment on approximation errors due to linear interpolation.** To measure the inter-frame detection performance of our method, we use linear interpolation between consecutive frames to generate ground truth. Although this linear interpolation affects ground truth accuracy within the interval because of interpolation errors, at the frame borders, that is, $t = 0$ ms and $t = 50$ ms, no approximation is made. Still, we verify the accuracy of the ground truth by evaluating our method for different interpolation methods. We focus on the subset that has object tracks that have a length of at least four and then apply cubic and linear interpolation of object tracks on the interval between the second and third frames. We report the results in Extended Data Fig. 3a. We see that the performance of our method deviates at most 0.2 mAP between linear and cubic interpolations. Although there is a small difference, we focus on using linear interpolation, as it allows us to use a larger subset of the test set for inter-frame object detection.

### Training details
On Gen1 and N-Caltech101, we use the AdamW optimizer[55] with a learning rate of 0.01 and weight decay of $10^{-5}$. We train each model for 150,000 iterations with a batch size of 64. We randomly crop the events to 75% of the full resolution and randomly translate them by up to 10% of the full resolution. We use the YOLOX loss[34], which includes an IOU loss, class loss and a regression loss, discussed in ref. 34. To stabilize training, we also use exponential model averaging[56].

On DSEC-Detection, we train with a batch size of 32, the learning rate of $2 \times 10^{-4}$ for 800 epochs using the AdamW optimizer[55], as before. Apart from the data augmentations described before, we now also use random horizontal flipping with a probability of 0.5 and random magnification with a scale $s \sim \mathcal{U}(1, 1.5)$. We train the network to predict with one image and 50 ms of events leading up to the next image, corresponding to the frequency of labels (20 Hz).

### Baselines
In the purely event-based setting, we compare with the following state-of-the-art methods.

**Dense recurrent methods.** In this category, RED (ref. 19) and ASTM-Net (ref. 28) are the state-of-the-art methods, and they feature recurrent architectures. We also include MatrixLSTM + YOLOv3 (ref. 29) that features a recurrent, learnable representation and a YOLOv3 detection head.

**Dense feedforward methods.** Reference 28 provides the results on Gen1 for the dense feedforward methods, which we term Events + RRC (ref. 38), Inception + SDD (ref. 26) and Events + YOLOv3 (ref. 27). These methods use dense event representations with the RRC, SSD or YOLOv3 detection head.

**Spiking methods.** We compare with the spiking network Spiking DenseNet (ref. 39), which uses an SSD detection head.

**Asynchronous methods.** Here we compare with the state-of-the-art methods AEGNN (ref. 31) and NVS-S (ref. 32), both graph-based, AsyNet (ref. 36), which uses submanifold sparse convolutions[57], and YOLE

(ref. 58), which uses an asynchronous CNN. All of these methods deploy their networks in an asynchronous mode during testing.

As implementation details are not available for Events + RRC (ref. 38), Inception + SDD (ref. 26) and Events + YOLOv3 (ref. 27), MatrixLSTM + YOLOv3 (ref. 29) and ASTM-Net (ref. 28), we find a lower bound on the per-event computation necessary to update their network based on the complexity of their detection backbone. Whereas for Events + YOLOv3 and MatrixLSTM + YOLOv3 we use the DarkNet-53 backbone, for ASTM-Net and Events + RRC, we use the VGG11 backbone, and for Inception + SDD the Inception v.2 backbone. As Spiking DenseNet uses spike-based computation, we do not report FLOPS because they are undefined and mark that entry with N/A.

**Hybrid methods.** In the event- and image-based setting, we additionally compare with an event- and frame-based baseline, which we term Events + YOLOX. It takes in concatenated images and event histograms[59] from events up to time $t$ and generates detections for time $t$.

**Image-based methods.** We compare with YOLOX (ref. 34). As YOLOX provides only detections at frame time, we present a variation that can provide detections in the blind time between the frames, using either constant or linear extrapolation of detections extracted at frame time. Whereas for constant extrapolation we simply keep object positions constant over time, for linear extrapolation we use detections in the past and current frames to fit a linear motion model on the position, height and width of the object. As YOLOX is an object detector, we need to establish associations between the past and current objects. We did this as follows: for each object in the current frame, we selected the object of the same class in the previous frame with the highest IOU overlap and used it to fit a linear function on the bounding box parameters (height, width, $x$ position and $y$ position). If no match was found (that is, all IOUs were 0 for the selected object), it was not extrapolated but instead kept constant.

Finally, we compare the bandwidth and latency requirements of the Prophesee Gen3 camera with those of a set of automotive cameras, which are summarized in Extended Data Table 2. We also illustrate the concept of bandwidth–latency trade-off in Fig. 1a. The bandwidth–latency trade-off, discussed in ref. 60, states that cameras such as the automotive cameras in Extended Data Table 2 cannot simultaneously achieve low bandwidth and low latency because of the reliance of a frame rate. By contrast, the Prophesee Gen3 camera can minimize both because it is an asynchronous sensor.

## Related work

**Dense neural network-based methods.** Since the introduction of powerful object detectors in classical image-based computer vision, such as R-CNN (refs. 61–63), SSD (ref. 64) and the YOLO series[34,65,66], and the widespread adoption of these methods in automotive settings, event-based object detection research has focused on leveraging the available models on dense, image-like event representations[19,26–29,38]. This approach enables the use of pretraining, and well-established architecture designs and loss functions, while maintaining the advantages of events, such as their high dynamic range, and negligible motion blur. Most recent examples of these methods include RED (ref. 19) and ASTM-Net (ref. 28), which operate recurrently on events and have shown high performance on detection tasks in automotive settings. However, owing to the nature of their method, these approaches necessarily need to convert events into dense frames. This invariably sacrifices the efficiency and high temporal resolution present in the events, which are important in many application scenarios such as low-power, always-on surveillance[67,68] and low-latency, low-power object detection and avoidance[3,69].

**Geometric learning methods.** As a result, a parallel line of research has emerged that tries to reintroduce sparsity into the present models by adopting either spiking neural network architectures[39] or geometric learning approaches[31,36]. Of these, spiking neural networks are capable of processing raw events asynchronously and are thus closest in spirit to the event-based data. However, these architectures lack efficient learning rules and thus do not yet scale to complex tasks and datasets[42,70–74]. Recently, geometric learning approaches have filled this gap. These approaches treat events as spatio-temporal point clouds[75], submanifolds[36] or graphs[31,32,43,76] and process them with specialized neural networks. Particular instances of these methods that have found use in large-scale point-cloud processing are PointNet++ (ref. 77) and Flex-Conv (ref. 78). These methods retain the spatio-temporal sparsity in the events and can be implemented recursively, in which single-event insertions are highly efficient.

**Asynchronous GNNs.** Of the geometric learning methods, processing events with GNNs is found to be most scalable, achieving high performance on complex tasks such as object recognition[32,43,44], object detection[31] and motion segmentation[45]. Recently, a line of research[31,32] has focused on converting these GNNs, once trained, into asynchronous models. These models can process in an event-by-event fashion while maintaining low computational complexity and generating an identical output to feedforward GNNs. They do so, by efficiently inserting events into the event graph[32], and then propagating the changes to lower layers, for which at each layer only a subset of nodes needs to be recomputed. However, these works are limited in three main aspects. First, they work only at a per node level, meaning that they flag nodes that have changed and then recompute the messages to recompute the feature of each node. This incurs redundant computation because effectively only a subset of messages passing to each changed node need to be recomputed. Second, they do not consider update pruning, which means that when node features do not change at a layer, they simply treat them as changed nodes, leading to additional computation. Finally, the number of changed nodes increases as the layer depth increases, meaning that these architectures work efficiently only for shallow neural networks, limiting the depth of the network.

In this work, we address all three limitations. First, we pass updates on a per-message level, that is, we recompute only messages that have changed. Second, we apply update pruning and explore a specialized network architecture that maximizes this effect by placing the max pooling layer early in the network. By modulating the number of output features of this layer, we can control the amount of pruning that takes place. Finally, we also apply a specialized LUT-SC that cuts the computation markedly. With the reduced computational complexity, we are able to design two times deeper architectures, which markedly boosts the network accuracy.

**Hybrid methods.** One of the reasons for the lower performance of event-based detectors also lies in the properties of the sensor itself. Although possessing the capability to detect objects fast and in high-speed and high-dynamic-range conditions, the lack of explicit texture information in the event stream prevents the networks from extracting rich semantic cues. For this reason, several methods have combined events and frames for moving-object detections[79], tracking[80], computational photography[22,81,82] and monocular depth estimation[40]. However, these are usually based on dense feedforward networks and simple event and image concatenation[22,82–84] or multi-branch feature fusion[40,83]. As events are treated as dense frames, these methods suffer from the same drawbacks as standard dense methods. In this work, we combine events and frames in a sparse way without sacrificing the low computational complexity of event-by-event processing. This is, to our knowledge, the first paper to address asynchronous processing in a hybrid network.

## Ablations

**Events only.** Here we motivate the use of the features of our method. We split our ablation studies into two parts: those targeting the efficiency

(Extended Data Fig. 4d) and those targeting the accuracy (Extended Data Fig. 4e) of the method. For all experiments, we use the model shown in Extended Data Fig. 1 without the image branch as a baseline and report the standard object detection score of mAP (higher is better)[85] on the validation set of the Gen1 dataset[41] as well as the computation necessary to process a single event in terms of floating point operations per event (FLOPS per event, lower is better).

*Ablations on efficiency.* Key building blocks of our method are LUT-SCs, which are an accelerated version of standard spline convolutions[35]. An enabling factor for using LUT-SCs lies in transitioning from 2D to 3D convolutions, which we investigate by training a model with 3D spline convolutions (Extended Data Fig. 4d, row 1). With an mAP of 31.84, it achieves a 0.05 lower mAP than our baseline (bottom row). Using 3D convolutions yields a slight decrease in accuracy and does not allow us to perform an efficient lookup, yielding 150.87 MFLOPS per new event. Using 2D convolutions (row 2) reduces the computation to 79.6 MFLOPS per event because of the dependence on the dimension $d$ in equation (12), which is further reduced to 17.3 MFLOPS per event after implementing LUT-SCs (row 3). In addition to the small increase in performance due to 2D convolutions, we gain a factor of 8.7 in terms of FLOPS per event.

Next, we investigate pruning. We recompute the FLOPS of the previous model by terminating update propagation after max pooling layers, shown in Extended Data Fig. 2b, and reported in Extended Data Fig. 4d (row 4). We find that this reduces the computational complexity from 17.3 to 16.3 MFLOPS per event. This reduction comes from removing the orange messages in Extended Data Fig. 2a (bottom). Implementing node position rounding in equation (6) (Extended Data Fig. 4d, row 5), enables us to fully prune updates. This method only requires 4.58 MFLOPS per event. Node position rounding reduces mAP only by 0.01, justifying its use.

In a final step, we also investigate the use of directed pooling, shown in Extended Data Fig. 2d. Owing to this pooling method, fewer edges are present after each pooling layer, thus restricting the message passing— that is, context aggregation abilities of our network. For this reason, it achieves only an mAP of 18.35. However, owing to the directedness of the graph, in each layer at most only one node needs to be updated (except for rare edge inversions), as shown in Extended Data Fig. 2c, leading to an overall computational complexity of only 0.31 MFLOPS per event. Owing to the lower performance, we instead use the previous method when comparing with the state-of-the-art methods. However, as will be seen later, the performance is affected to a much lesser degree when combined with images.

*Ablations on accuracy.* We found that three features of our network had a marked impact on performance. First, we applied early temporal aggregation, that is, using $g_t = 1$, which sped up training and led to higher accuracy. We trained another model that pooled the temporal dimension more gradually by setting $g_t = 8/2^i$, where $i$ is the index of the pooling layer. This model reached only an mAP of 21.2 (Extended Data Fig. 4e, row 3), after reducing the learning rate to 0.002 to enable stable training. This highlights that early pooling plays an important part because it improves our result by 10.6 mAP. We believe that it is important for mixing features quickly so that they can be used in lower layers.

Next, we investigate the importance of network depth on task performance. To see this, we trained another network, in which we removed the skip connection and second (LUT-SC and BN) block from the layer in Extended Data Fig. 1c, which resulted in a network with a total of eight layers, on par with the network in ref. 31, which had seven layers. We see that this network achieves only an mAP of 22.5 (Extended Data Fig. 4e, row 2) highlighting the fact that 9.4% in mAP is explained by a deeper network architecture. We also combine this ablation with the previous one about early pooling and see that the network achieves only 15.8 mAP, another drop of 6.7% mAP (Extended Data Fig. 4e, row 1). This result is on par with the result in ref. 31, which achieved a performance of 16.3 mAP, on par with our method. This highlights the importance of using a deep neural network to boost performance.

Finally, we investigate using multiple layers before the max pooling layer. We train another model that only has a single-input layer, replacing the layer in Extended Data Fig. 1 with a (LUT-SC, BN and ReLU) block. This yielded a performance of 30.0 mAP (Extended Data Fig. 4e, row 4), which is 1.8 mAP lower than the baseline (Extended Data Fig. 4e, row 5). The computational complexity is only marginally lower, which is explained by Extended Data Fig. 2c (top). We see that adding layers at the input generates only a few additional messages. This highlights the benefits of using a directed event graph.

*Timing experiments.* We compare the time it takes for our dense GNN to process a batch of 50,000 events averaged over Gen1, and compare it with our asynchronous implementation on a Quadro RTX 4000 laptop GPU. We found that our dense network takes 30.8 ms, whereas the asynchronous method requires 8.46 ms, a 3.7-fold reduction. We believe that with further optimizations, and when deployed on potentially spiking hardware, this method can reduce power and latency by additional factors.

**Max pooling.** In this section, we take a closer look at the pruning mechanism. We find that almost all pruning happens in the very first max pooling layer. This motivates the placement of the pooling layer at the early stages of the network, which allows us to skip most computations when pruning happens. Also, as the subgraph is still small in the early layers, it is easy to prune the entire update tree. We interpret this case as event filtering and investigate this filter in Extended Data Fig. 4.

When applied to raw events (Extended Data Fig. 4a), we obtain filtered events (Extended Data Fig. 4b), that is, events that passed through the first max pooling layer. We observe that max pooling makes the events more uniformly distributed over the image plane. This is also supported by the density plot in Extended Data Fig. 4b, which shows that the distribution of the number of events per-pixel shifts to the left after filtering, removing events in regions in which there are too many. This behaviour can be explained by the pigeon-hole principle when applied to max pooling layers. Max pooling usually uses only a fraction of its input nodes to compute the output feature. The number of input nodes used by the max pooling layer is upper bounded by its output channel dimension, $c_{out}$, because it could at maximum use only one feature from each input node. As a result, max pooling selects at most $c_{out}$ nodes for each voxel, resulting in more uniformly sampled events.

To study the effect of the output channel dimension on filtering, we train four models with $c_{out} \in \{8, 16, 24, 32\}$, in which our baseline model had $c_{out} = 16$. We report the mAP, MFLOPS per event and fraction of events after filtering, $\phi$ averaged over Gen1, in Extended Data Fig. 4c. As predicted, we find that increasing $c_{out}$ increases mAP, MFLOPS and $\phi$. However, the increase happens at different rates. While MFLOPS and $\phi$ grow roughly linearly, mAP growth slows down significantly after $c = 24$. Interestingly, by selecting $c_{out} = 8$ we still achieve an mAP of 30.6, while using only 21% of events. This type of filtering has interesting implications for future work. An interesting question would be whether events that are not pruned carry salient and interpretable information.

**Images and events.** In this section, we ablate the importance of different design choices when combining events and images. In all experiments, we report the mAP and mean number of MFLOPS per newly inserted event over the DSEC-Detection validation set. When computing the FLOPS, we do not take into account the computation necessary by the CNN, because it needs to be executed only once. Our baseline model uses DAGr-S for the events branch and ResNet-18 (ref. 30).

*Ablations on fusion.* In the following ablation studies, we investigate the influence of (1) the feature sampling layer and (2) the effect on detection

adding at the detection outputs of an event and image branch. We summarize the results of this experiment in Extended Data Fig. 5d. In summary, we see that our baseline (Extended Data Fig. 5d, row 4) achieves an mAP of 37.3 with 6.73 MFLOPS per event. Removing feature sampling results in a drop of 3.1 mAP, while reducing the computational complexity by 0.73 MFLOPS per event. We argue that the performance gain due to feature sampling justifies this small increase in computational complexity. Removing detection adding at the output reduces the performance by 5.8 mAP, while also reducing the computation by 1.24 MFLOPS per event. We argue that this reduction comes from the fact that the image features are predominantly used to generate the output (that is, compared with the events only, which is 18.5 mAP lower), and thus more event features are pruned at the max pooling layer (roughly 20% more). Finally, if both feature sampling and detection adding are removed, we arrive at the original DAGr architecture, which achieves an mAP of 14.0 with 6.05 MFLOPS per event. It has a computational complexity on par with the baseline with detection adding, but with a performance of 20.2 mAP lower, justifying the use of detection adding.

*Other ablations.* We found that two more factors helped the performance of the method without affecting the computation markedly: (1) CNN pretraining and (2) concatenation of image and event features that we ablate in Extended Data Fig. 5e. To test the first feature, we train the model end to end, without pretraining the CNN branch, and found that it resulted in a 0.2-mAP reduction in performance, with a negligible reduction in computational complexity. Next, we replaced the concatenation operation with a summation, which reduces the number of input channels to each spline convolution. This change reduces the mAP by 0.5 mAP and the computation by 1.24 MFLOPS per event. Instead, naive concatenation requires 7.49 MFLOPS per event without the simplifications in equation (18). If we use equation (18), we can reduce this computation to 6.74 MFLOPS per event, a roughly 10% reduction with no performance impact.

*Ablation on CNN backbone.* We evaluate the ability of our method to perform inter-frame detection using different network backbones, namely, ResNet-18, ResNet-34 and ResNet-50, and provide the results in Extended Data Fig. 5a. Green and reddish colours indicate with and without events, respectively. As seen previously with the ResNet-50 backbone event and image-based methods (green), all show stable performance, successfully detecting objects in the 50 ms between two frames. As the backbone capacity increases, their performance level also increases. We also observe that with increasing time $t$ ranging from 0 ms to 50 ms, all methods slightly increase, reach a maximum and then decrease again, improving the initial score at $t = 0$ by between 0.6 mAP and 0.7 mAP. The performance increase can be explained because of the addition of events, that is, more information becomes available so that detections can be refined, especially in the dark and blurry regions of the image. The subsequent slight decrease can then be explained by the fact that image information becomes more outdated. By contrast, purely image-based methods (red) suffer significantly in this setting. While starting off at the same level as the image and event-based methods, they quickly degrade by between 8.7 mAP and 10.8 mAP after 50 ms. The performance change over time for all methods is shown in Extended Data Fig. 5c, in which we confirm our findings. This decrease highlights the importance of updating the prediction between the frames. Using events is an effective and computationally cheap way to do so, closing the gap of up to 10.8 mAP. We illustrate this gain in performance by using events qualitatively in Fig. 5, in which we show object detections of DAGr-S + ResNet-50 in edge-case scenarios.

*Timing experiments.* We report the runtime of our method in Extended Data Table 1 and find the fastest method to be DAGr-S + ResNet50 with 9.6 ms. Specific hardware implementations are likely to reduce this number substantially. Moreover, as can be seen in the comparison, MFLOPS per event does not correlate with runtime at these low computation regimes, and this indicates that significant overhead is present in the implementation. We use the PyTorch Geometric[86] library, which is optimized for batch processing, and thus introduces data handling overhead. When eliminating this overhead, runtimes are expected to decrease even more.

#### Further experiments on DSEC-Detection

**Event cameras provide additional information.** One of the proposed use cases for an event camera is to detect objects before they become fully visible within the frame. These could be objects, or parts of objects, appearing from behind occlusions, or entering the field of view. In this case, the first image does not carry sufficient information to make an informed decision, which requires waiting for information from additional sensors, or integrating context-enriched information from details such as shadows and body parts. Integrating this information can reduce the uncertainties in partially observable situations and is applicable to both image- and event-based algorithms. Event cameras, however, provide additional information, which invariably enhances prediction, even under partial observability (for example, an arm appearing from behind an occlusion or a cargo being lost on a highway). To test this hypothesis, we compared our method with the image-based baseline with extrapolation on the subset of DSEC-Detection in which objects suddenly appear or disappear (a total of 8% of objects). This subset requires further information to fill in these detections. Our event- and image-based method achieves 37.2 mAP, and the image-based method achieves 33.8 mAP, showing that events can provide a 3.4-mAP boost in this case.

**Incorporating CNN latency into the prediction.** Our hybrid method relies on dense features provided by a standard CNN, which is computationally expensive to run. We thus try to understand if our method would also work in a scenario in which dense features appear only after computation is finished and then need to be updated by later events. To test this case, we perform the following modification to our method. After a computation time $\Delta t$ for computing the dense features, we integrate the events from the interval $[\Delta t, 50$ ms$]$ into the detector. This means that for time $0 < t < \Delta t$, no detection can be made, as no features are available from images. In this interval, either the event-only method from Extended Data Fig. 5d (row 1) can be used, or a linear propagation from the detection from the previous interval. At time $t > \Delta t$, we use the events in interval $[\Delta t, t]$. The runtimes for the different image networks (ResNet-18, ResNet-34 and ResNet-50 + detection head) were 5.3 ms, 8.2 ms and 12.7 ms, respectively, on a Quadro RTX 4000 laptop GPU. We report the results in Extended Data Fig. 3c. We see that on the full DSEC-Detection test set after 50-ms events, DAGr-S + ResNet-50 achieves a performance of 41.6 mAP, 0.3 mAP lower than without latency consideration. On the inter-frame detection task, this translates to a reduction from 44.2 mAP to 43.8 mAP, still 6.7 mAP higher than the image-based baseline with extrapolation implemented. This demonstrates that our method outperforms image-based methods even when considering computational latency due to CNN processing. For smaller networks ResNet-34 and ResNet-18, the degradations on the full test set are 0.1 mAP and 0.1 mAP, respectively, compared with the corresponding methods without latency consideration. Notably, smaller networks have lower latency and thus incur smaller degradations. However, the largest model still achieves the highest performance. Nonetheless, to minimize the effect of this latency, future work could consider incorporating the latency into the training loop, in which case the method will probably learn to compensate for it.

#### Research ethics

The study has been conducted in accordance with the Declaration of Helsinki. The study protocol is exempt from review by an ethics committee according to the rules and regulations of the University of Zurich, because no health-related data have been collected. The participants gave their written informed consent before participating in the study.

## Data availability
The data that support the findings of this study are available from the corresponding authors upon reasonable request.

## Code availability
The open-source code can be found at GitHub (https://github.com/uzh-rpg/dagr) and the instructions to download the DSEC-Detection can be found at GitHub (https://github.com/uzh-rpg/dsec-det).

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

**Acknowledgements** We thank M. Gehrig, M. Muglikar and N. Messikommer, who contributed to the curation of DSEC-Detection, and R. Sabzevari for providing insights and comments. This work was supported by Huawei Zurich, the Swiss National Science Foundation through the National Centre of Competence in Research (NCCR) Robotics (grant no. 51NF40_185543) and the European Research Council (ERC) under grant agreement no. 864042 (AGILEFLIGHT).

**Author contributions** D.G. formulated the main ideas, implemented the system, performed the experiments and data analysis, and wrote the paper; D.S. contributed to the main ideas, the experimental design, analysis of the experiments, writing of the paper, and provided funding.

**Funding** Open access funding provided by University of Zurich.

**Competing interests** The authors declare no competing interests.

**Additional information**
**Correspondence and requests for materials** should be addressed to Daniel Gehrig or Davide Scaramuzza.

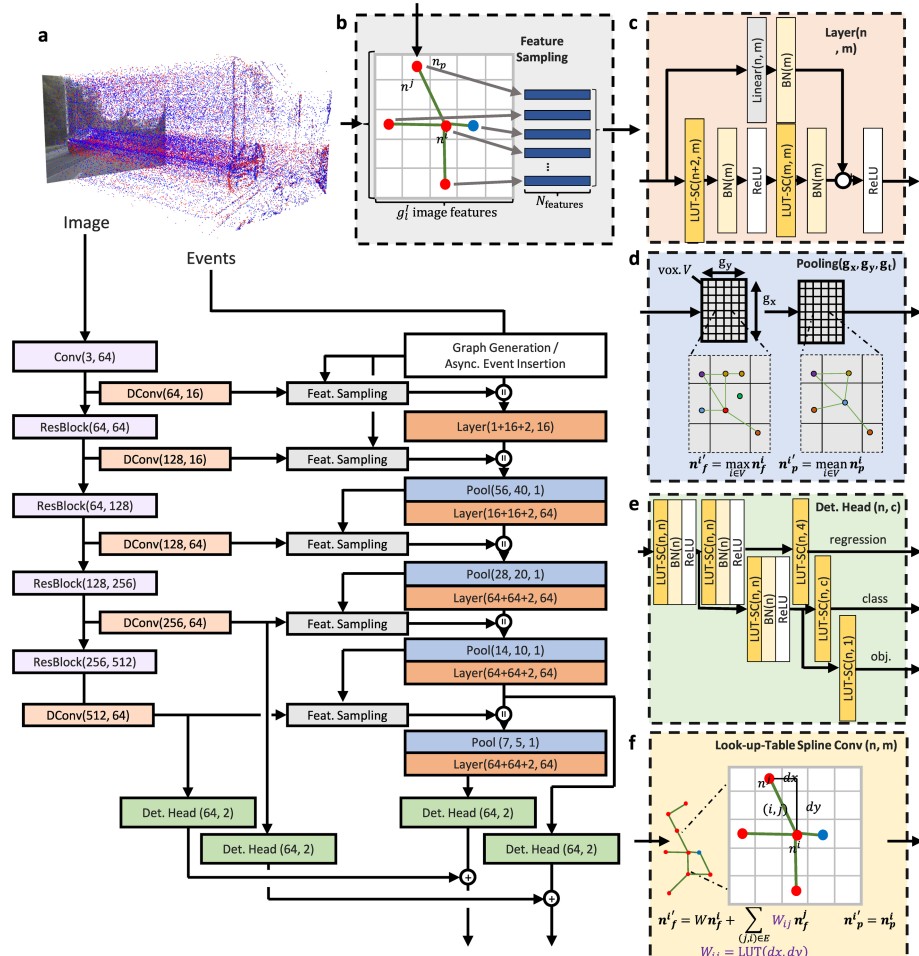

**Extended Data Fig. 1 | Overview of the network architecture of DAGr.**
(a) General architecture overview showing the CNN-based ResNet-18[30] branch and the GNN. Each sensor modality is processed separately, while sharing features, and adding objectness, classification and regression scores at the output. (b) Directed feature sampling layer. Graph nodes sample features at the corresponding pixel locations and concatenate them with their own feature. (c) Residual blocks, with arguments $n$ and $m$ denoting input an output channels

dimension. The + 2 means concatenation with the 2D node position. (d) Max pooling layer with arguments $g_x$, $g_y$ and $g_t$ denoting the number of grid cells in each dimension. (e) Multiscale YOLOX-inspired detection head, outputting bounding boxes (*regression*), class scores and object confidence. (f) Look-up-Table Spline Convolution (LUT-SC), which use uses discrete-valued relative distance between neighboring nodes to look up a weight matrices.

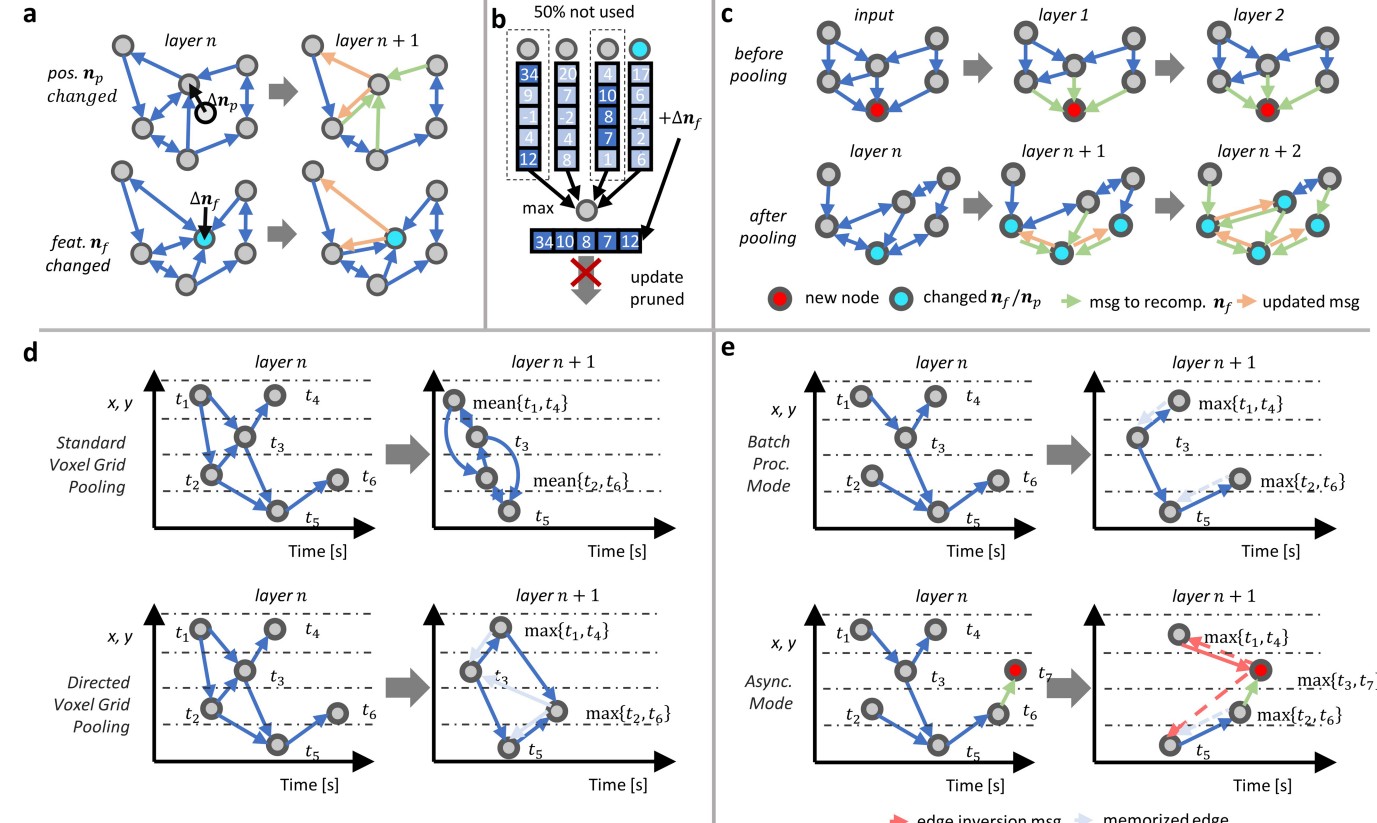

**Extended Data Fig. 2 | Asynchronous graph operations for a single event.**
(a) Update rule for convolution layers. Node position or feature changes result in update messages from the changed node (orange arrows). Node position changes result in recompute messages to the changed node (green messages). (b) Update pruning in pooling layers. If a changed input node s is in the currently unused (grayed out) set, it does not have a feature higher than the current output and it does not change the output node sufficiently to change rounding, the update is pruned. (c) Update propagation applied to multiple layers. Before pooling, edges are directed, so the number of computed messages remains constant with network depth. After pooling, bidirectional edges appear,

leading to a growth in the number of computed messages in lower layers. (d) To reduce this growth, directed voxel grid pooling is introduced. Different to standard pooling, directed pooling max-pools over the time dimension, and filters edges for which the source node has a higher timestamp than the input node (grayed out), resulting in a directed event graph even after pooling. (e) Asynchronous updating of directed pooling layer. Sometimes edges are inverted when an older node is promoted to a newer node through max-pooling of the time dimension. In this case, the edges need to be reversed, leading to a new message (pink) being sent to and from the updated node.

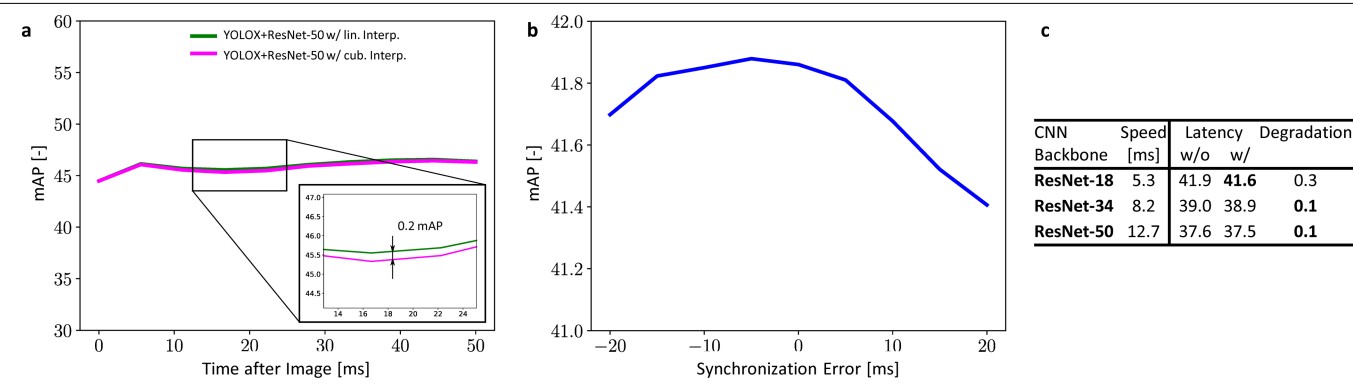

**Extended Data Fig. 3 | Sensitivity analysis of our method.** (a) Performance of DAGr+ResNet-50 on the DSEC-Detection test-set for different event-to-image synchronization errors. (b) Object detection score of our method for differently interpolated ground truth. Here we use the DSEC-Detection subset with object tracks with a length of at least four. (c) Performance of different CNN backbones with DAGr-S on the full DSEC-Detection test set, with and without CNN latency considered. All performances are measured in mAP (higher is better). Speed refers to the average runtime of the CNN alone over images from the test set.

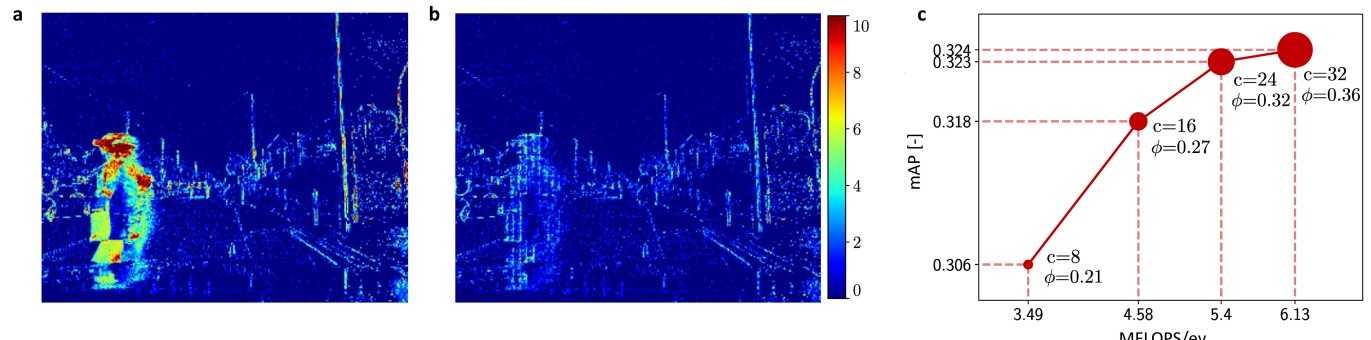

**d**

| 2D conv | LUT-SC | pruning | pos. rounding | directed pooling | mAP↑ | MFLOPS/ev↓ |
|---|---|---|---|---|---|---|
| ✗ | ✗ | ✗ | ✗ | ✗ | 31.84 | 150.87 |
| ✓ | ✗ | ✗ | ✗ | ✗ | 31.90 | 79.6 |
| ✓ | ✓ | ✗ | ✗ | ✗ | 31.90 | 17.3 |
| ✓ | ✓ | ✓ | ✗ | ✗ | **31.90** | 16.3 |
| ✓ | ✓ | ✓ | ✓ | ✗ | 31.79 | 4.58 |
| ✓ | ✓ | ✓ | ✓ | ✓ | 18.35 | **0.311** |

**e**

| early aggregation | multi-layer input | deep network | mAP↑ | MFLOPS/ev↓ |
|---|---|---|---|---|
| ✗ | ✗ | ✗ | 15.8 | 2.02 |
| ✓ | ✗ | ✗ | 22.5 | 1.94 |
| ✗ | ✓ | ✓ | 21.2 | 6.27 |
| ✓ | ✗ | ✓ | 30.0 | 4.56 |
| ✓ | ✓ | ✓ | **31.8** | 4.58 |

**Extended Data Fig. 4 | Ablations on different network components of the GNN.** (a-d) Effect of update pruning due to max pooling. We interpret max pooling as a kind of event filter. In (a-b) we show an example of aggregated events before (a) and after (b) filtering. This filter acts as a saliency detector, only letting through events with "new information", and removing redundant events in high event rate regions. This results in a more uniform distribution of events (c). We can control the filter strength by modulating the number of output features, $c$. As seen in (d), increasing $c$ increases both computation and mAP. However, mAP growth drastically reduces in slope after $c = 24$. The dot size is proportional to $c$, and $\phi$ measures the proportion of updates that pass through the filter. In our baseline setting with $c = 16$, we see that only 27% of updates pass the first max pooling layer. (e) Features affecting computational complexity. (f) Features affecting accuracy.

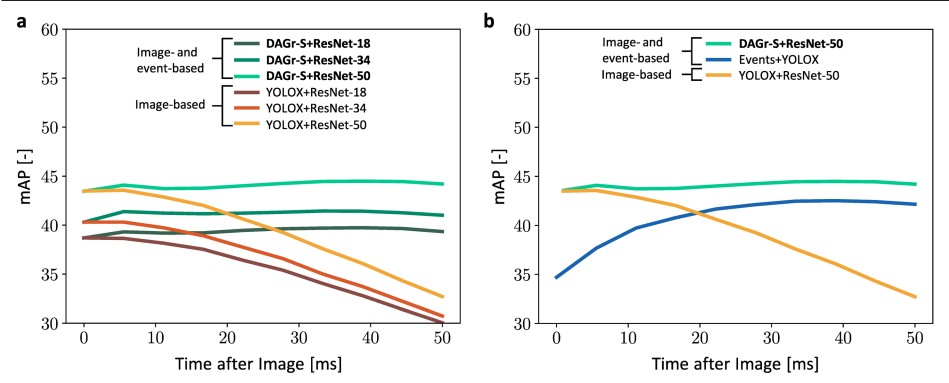

**c**

| Method | uses events | mAP↑ | | | |
|---|---|---|---|---|---|
| | | 0 ms | max | 50 ms | Δ |
| Events+YOLOX | ✓ | 34.7 | 42.5 | 42.2 | +7.5 |
| YOLOX+ResNet-18 | ✗ | 38.7 | 38.7 | 30.0 | -8.7 |
| YOLOX+ResNet-34 | ✗ | 40.3 | 40.3 | 30.7 | -9.6 |
| YOLOX+ResNet-50 | ✗ | 43.5 | 43.5 | 32.7 | -10.8 |
| **DAGR-S+ResNet-18** | ✓ | 38.7 | 39.7 | 39.3 | +0.6 |
| **DAGR-S+ResNet-34** | ✓ | 40.3 | 41.4 | 41.0 | +0.7 |
| **DAGR-S+ResNet-50** | ✓ | 43.5 | 44.5 | 44.2 | +0.7 |

**d**

| Input modality | feature sampling | detection adding | mAP | MFLOPS/ev |
|---|---|---|---|---|
| events | ✗ | ✗ | 14.0 | 6.05 |
| events+images | ✓ | ✗ | 32.5 | **5.49** |
| events+images | ✗ | ✓ | 34.2 | 6.00 |
| events+images | ✓ | ✓ | **37.3** | 6.73 |

**e**

| Input modality | mAP | MFLOPS/ev |
|---|---|---|
| Baseline | 37.3 | 6.73 |
| w/o CNN pretraining | 37.1 | 6.62 |
| w/o concatenation | 36.8 | **5.87** |
| w/o concat. simplification | **37.3** | 7.49 |

**Extended Data Fig. 5 | Ablations on components of the hybrid network.**
(a-c) Compare the network performance between two frames, on a subset of
DSEC-Detection. (a) Our methods performance with ResNet-18, ResNet-34, and
ResNet-50 backbones, with events (green), and without events (red). Methods
without events propagate detections from the image at $t = 0$ to the current
time. (b) Comparison of our method to Events+YOLOX[34] (blue), a baseline
which takes in concatenated images and events up to time $t$. (c) Drop in mean
average precision (mAP) over time, for each method. (d-c) Compare the
network performance on the full DSEC-Detection test set. (d) Ablation on the
fusion strategies between GNN-based detections from events and CNN-based
detections from images. (e) Ablation on CNN pretraining and feature
concatenation.

**a**

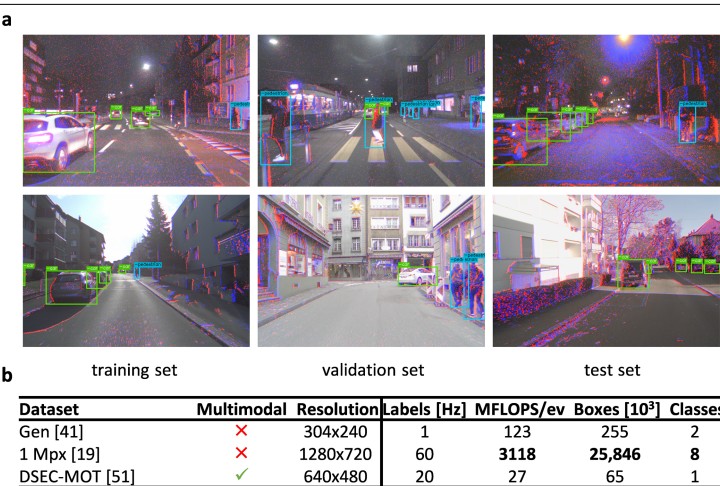

| training set | validation set | test set |
| :---: | :---: | :---: |

**b**

| Dataset | Multimodal | Resolution | Labels [Hz] | MFLOPS/ev | Boxes [$10^3$] | Classes |
| :--- | :---: | :---: | :---: | :---: | :---: | :---: |
| Gen [41] | ✗ | 304x240 | 1 | 123 | 255 | 2 |
| 1 Mpx [19] | ✗ | 1280x720 | 60 | **3118** | **25,846** | **8** |
| DSEC-MOT [51] | ✓ | 640x480 | 20 | 27 | 65 | 1 |
| **DSEC-Detection (ours)** | ✓ | 640x480 | **20** | 70 | 390 | 8 |

**Extended Data Fig. 6 | Preview of DSEC-Detection.** (a-c) Preview of samples from the training (a), validation (b) and test set (c) of DSEC-Detection. It features spatiotemporally aligned events and frames with object detection labels for pedestrians and cars. (d) Breakdown of the data in DSEC-Detection, and comparison with related work.

## Extended Data Table 1 | Quantitative comparison of our method against state-of-the-art

**a**

| Method | Async. | Gen1 | | | N-Caltech101 | | |
|---|---|---|---|---|---|---|---|
| | | mAP↑ | MFLOPS/ev↓ | µJ/ev↓ | mAP↑ | MFLOPS/ev↓ | µJ/ev↓ |
| Inception+SSD [26] | ✗ | 30.1 | 27,183 | 23,502 | - | - | - |
| Events+RRC [38] | ✗ | 30.7 | >21,758* | >18,386 | - | - | - |
| MatrixLSTM+YOLOv3 [29] | ✗ | 31.0 | >34,519* | >29,168 | - | - | - |
| Events+YOLOv3 [27] | ✗ | 31.2 | 65,558 | 55,397 | - | - | - |
| RED [19] | ✗ | 40.0 | 4,712 | 3,982 | - | - | - |
| ASTM-Net [28] | ✗ | 46.7 | >21,758* | >18,386 | - | - | - |
| NVS-S [32] | ✓ | 8.60 | 7.80 | 6.59 | 34.6 | 7.80 | 6.59 |
| AsyNet [36] | ✓ | 14.5 | 205 | 173 | 64.3 | 200 | 169 |
| AEGNN [31] | ✓ | 16.3 | 5.26 | 4.44 | 59.5 | 7.41 | 6.26 |
| Spiking DenseNet [39] | ✓ | 18.9 | N/A | - | - | - | - |
| YOLE [57] | ✓ | - | - | - | 39.8 | 3682 | 3111 |
| **DAGr-N** | ✓ | 26.3 | **1.36** | **1.15** | 62.9 | **2.28** | **1.93** |
| **DAGr-S** | ✓ | 30.4 | 4.58 | 3.87 | 70.2 | 6.85 | 5.76 |
| **DAGr-M** | ✓ | 31.8 | 9.94 | 8.40 | 72.7 | 12.2 | 10.6 |
| **DAGr-L** | ✓ | **32.1** | 17.4 | 14.7 | **73.2** | 18.9 | 16.0 |

* Lower bound from network backbone

N/A: FLOPS are undefined due to spike-based computation

**b**

| Method | Async. | DSEC-Detection | | | Speed [ms] |
|---|---|---|---|---|---|
| | | mAP↑ | MFLOPS/ev↓ | µJ/ev↓ | |
| Inception +SSD [26] | ✗ | 18.4 | 27,183 | 23,501 | - |
| Events+YOLOv3 [27] | ✗ | 28.7 | 65,558 | 55,396 | - |
| Events+YOLOX [34] | ✗ | 40.2 | 22,049 | 18,631 | - |
| **DAGr-S+ResNet-18** | ✓ | 37.6 | 6.74 | 5.70 | **10.05±1.68** |
| **DAGr-S+ResNet-34** | ✓ | 39.0 | 6.57 | 5.55 | 10.56±1.83 |
| **DAGr-S+ResNet-50** | ✓ | **41.9** | **6.41** | **5.42** | 9.64±1.47 |
| **DAGr-S+ResNet-18*** | ✓ | 35.8 | 0.572 | 0.967 | **11.79±1.79** |
| **DAGr-S+ResNet-34*** | ✓ | 37.6 | 0.570 | 0.963 | 12.14±1.79 |
| **DAGr-S+ResNet-50*** | ✓ | 39.9 | **0.569** | 0.961 | 12.43±1.83 |

*indicates with directed voxel pooling

(a) Comparison against asynchronous methods in terms of computational complexity, task performance, and energy consumption. Results of event-based methods on the Gen1 detection dataset[41] and N-Caltech101[42]. (b) Comparison of event and image-based detectors on DSEC-Detection. Here, methods are tasked to predict labels 50 ms after the first image, given a single image, and events. Speed refers to the average runtime of event insertion into the GNN averaged over the dataset.

**Extended Data Table 2 | Overview of the automotive cameras that are currently in use**

| Company | Sensor | Res. [MP] | FPS | Latency [ms] |
|---|---|---|---|---|
| Cruise [4] | - | 0.7 | 30 | 33.3 |
| MobileEye [9] | KAC-9619 | 8.0 | 30 | 33.3 |
| Omnivision[8] | OX08B | 8.3 | 36 | 27.8 |
| Tesla [5] | IMX490 | 5.4 | 40 | 25.0 |
| Bosch [7] | MPC3 | 2.6 | 45 | 22.2 |
| Sony [6] | IMX290NQV | 2.1 | 120 | 8.33 |
| Sony [6] | IMX224 | 1.3 | 120 | 8.33 |
| Prophesee [86] | Gen 3.1 | 0.3 | - | **0.2** |

For each camera we provide the resolution in megapixels (MP), frames per second (FPS) and perceptual latency, in milliseconds. Data from refs. 4–9,52.