## [Peer Review File · Nature]

Manuscript Title: Low Latency Automotive Vision with Event Cameras

Reviewer Comments & Author Rebuttals

Reviewer Reports on the Initial Version:

Referee #1 (Remarks to the Author):

With their work and with this submission, the authors address one of the most important design aspects for the next generation of driver assistance systems and autonomous vehicles - that of ensuring minimal information gaps in sensing the environment and understanding the traffic scene in order to enable safe and reliable decisions. Today's vision-based perception systems - key building blocks in the whole chain - are data-intensive, posing huge computational and communication challenges and driving up material costs significantly. Reliable solutions that significantly reduce these burdens would be highly sought after by engineers and researchers.

Novelty in approach:

The authors present a method for combining low-frequency RGB images from conventional fixed-frame-rate cameras with high-frequency single event information from event-triggered cameras, in order to continue to generate object detections during the 'blind' periods between two conventional camera frames. The methods used to adapt and combine the advantages of a convolutional neural network - in this case tasked with extracting the object features - with those of a graph neural network to handle the event information are indeed novel, innovative and appropriate for the stated goal. The implementation of a series of modifications to the CNN and subsequent GNN, aimed at improving their training and inference efficiency, although common, provide valuable insights in a new scenario.

Structure and presentation:

Overall, the article is very well structured and presented. In support of their proposed methods, the authors have provided a comprehensive and plausible overview comparing their approach with the state of the art and have presented numerous combinatorial ablation studies to support their conclusions.

Weaknesses:

The main weakness is the lack of a sound motivation for the work from an application point of view. Why is it necessary to use event cameras to fill the information gap between camera frames? This should be elaborated in the context of the intended use of such combined systems.

The authors argue for the use of high-frequency but sparse event information in the following situations to improve object detection and positioning:

1. When an object enters the field of view, frame-based detectors may not detect the object immediately because not all of its distinguishing features are fully visible.

2. When an object is partially obscured by other objects.

3. Positioning an object between two successive camera frames can only be done if additional information is available.

In this context, we suggest that the authors consider the following possible counter-arguments:

Case 1 above could possibly be addressed by training the detectors with context-enriched data, e.g. by looking for outstretched arms and legs of pedestrians and predicting their presence. While the uncertainties in detection and classification could increase in such cases, plausibility checks based on additional contextual information from high-resolution maps can help to reduce the uncertainties to an acceptable level.

For cases 2. and 3. above, it may also be possible to fill the information gap by extrapolating previous camera detections based on either a simple linear model or, where possible, an object-class-specific realistic motion model.

The authors already use an interpolation strategy to generate the ground truth bounding boxes for image time stamps between two recorded frames. The authors may wish to consider using a similar approach directly to generate in-between data.

It may also be the case that extrapolation, which is not a new certified snapshot of reality, could lead to larger errors - in which case it would be more appropriate to use the event information. Which one to use in which situation would also be an interesting question.

Evaluations:

The authors used YoloX as the base model for comparisons. As the newer models such as YoloV7 or YoloV8 outperform YoloX by at least 3% for mean average precision, the authors' comments on whether this would affect the comparisons would be highly appreciated.

Additional aspects which should be addressed in the opinion of the reviewer:

- Handling of stationary and relatively stationary objects
- Statistics for edge cases
- Time synchronisation requirements between camera frames and events
- Communication network implications and event transport delays

Referee #2 (Remarks to the Author):

A. the authors propose a method to fuse RGB images and events from event-cameras for object detection. The results shows the increase of mean average precision with respect to object detectors based on RGB or events singularly, in particular during interframe detections, while keeping the bandwidth and latency low.

For RGB they use resnet-like architectures and for events they use a asynchronous graph CNN called Deep Asynchronous Graph Neural Network (DAGr). They also propose strategies such as pruning to reduce computation on Graph CNN.

B. The paper is original in the sensor fusion aspect combining state of the art methods for RGB object detection and event-based object detection, and novel in the strategies to improve the performance of the DAGr.

C. the authors use available datasets and created a dataset that combines event and rgb data. The ground truth (GT) boxes positions is approximated linearly between frames, which introduce small errors. An another source of error is the alignment based on an homography between frame and event cameras.

Nonetheless, the quality of the data seems sufficient for the task, that does not require a pixel-level precision as described. If the system is used for decision making like in an autonomous car more precise datasets might be required.

The approach is sound and the presentation is good with several graphs.

D. The statistical uncertainty of the results is not described, even though is rarely provided in similar works.

E. The conclusions are correct highlighting the advantage of the proposed approach and also the limitation on standard hardware. The conclusions are based on several ablation studies therefore they seems robust and reliable.

F. I would define the perceptual and computational latency to avoid confusion. It would be interesting also to consider a real scenario, where the dense features are obtained only after the frame rgb acquisition and after dense computation. Therefore the dense features are outdated by a certain latency. It is not clear whether such realistic scenario has been considered and if there are implication in the results.

It would also be interesting to have time performance and not just Mflops in the various tables. Obviously, the time is hardware dependent, but it gives an idea, especially on how much parallelization is possible (MFlops do not give an idea on how many of such flops can be done in parallel).

typos and remarks:

page 5: the "s" seems a typo: FLIR BlackFly s global shutter,

page 5: in this section is not entirely clear that the performance is compared (I assume) over time:

"We also observe that all methods slightly increase, reach a maximum, and then decrease again, improving the initial score at $t = 0$ by between 0.6 and 0.7 mAP. The performance increase can be

explained due to the addition of events". I suggest to make it more explicit.
page 11: double dots: high accuracy and efficiency..

G. the references are appropriate and they properly credit previous work for both event and frame-based data. If possible I would add some reference to other point cloud methods, for example Flex-convolution and PointNet. They might be relevant for the related work.

PointNet++: Deep Hierarchical Feature Learning on Point Sets in a Metric Space Hao Su, Charles R. Qi, Li Yi, Leonidas J. Guibas

Flex-Convolution (Million-Scale Point-Cloud Learning Beyond Grid-Worlds) Fabian Groh, Patrick Wieschollek, Hendrik P.A. Lensch

H. I would make the contributions more explicit in the introduction, e.g. the pruning. The rest of the text is overall clear.

Author Rebuttals to Initial Comments:

Comments by Referee #1

1. The main weakness is the lack of a sound motivation for the work from an application point of view. Why is it necessary to use event cameras to fill the information gap between camera frames? This should be elaborated in the context of the intended use of such combined systems.

The authors argue for the use of high-frequency but sparse event information in the following situations to improve object detection and positioning:

1. When an object enters the field of view, frame-based detectors may not detect the object immediately because not all of its distinguishing features are fully visible.
2. When an object is partially obscured by other objects.
3. Positioning an object between two successive camera frames can only be done if additional information is available.

In this context, we suggest that the authors consider the following possible counter-arguments:

Case 1 above could possibly be addressed by training the detectors with context-enriched data, e.g. by looking for outstretched arms and legs of pedestrians and predicting their presence.

Using context-enriched data, as outlined by the reviewer to address case 1, seeks to enhance the prediction under *partial observability*, and is thus applicable to both event and image data. This makes it an orthogonal design choice, likely to improve both image-based and event-based methods, but, crucially, incapable of generating certifiable snapshots of the real world due to an absence of additional information. Event cameras provide this additional information, which invariably enhances prediction, even under *partial observability* (e.g. an arm appearing from behind an occlusion, or cargo being lost on a highway). To test this hypothesis we compared our method against an image-based baseline with extrapolation (as suggested by the reviewer in comment 4, and explained in more detail there) on the subset of DSEC-Det in which objects suddenly *appear or disappear* (a total of 8% of objects). This subset requires additional information to fill in these detections. Our event and image-based method achieves 37.2 mAP, and the image-based method achieves 33.8 mAP. This shows that events add much needed information that provides a 3.4 mAP boost for case 1.

We summarize this experiment in the supplement Sec. 8 “Event cameras provide additional information”.

2. While the uncertainties in detection and classification could increase in such cases, plausibility checks based on additional contextual information from high-resolution maps can help to reduce the uncertainties to an acceptable level.

We agree that using additional contextual information can reduce uncertainties but likely not to an acceptable level. This is because this information, especially a single snapshot of e.g. outstretched arms lacks *motion information*. This motion information is an important indicator of intent and the planned trajectories of cars and pedestrians, and can likely only be gathered through a sequence of images which increases the latency. In fact, in motion forecasting methods like [A, B, C] which are becoming part of commercial patents [D], this latency is on the order of 0.5-2.0 seconds. Event data carries this motion information explicitly in the form of the slope in the spatio-temporal domain and can thus bridge this gap, and reduce uncertainties to acceptable levels.

We include this argument in Sec. 1.

[A] Alexander Cui, Sergio Casas, Kelvin Wong, Simon Suo, Raquel Urtasun, “GoRela: Go Relative for Viewpoint-Invariant Motion Forecasting”, ICRA 2023.

[B] Zikang Zhou, JianPing Wang, Yung-Hui Li, Yu-Kai Huang, “Query-Centric Trajectory Prediction”, CVPR 2023

[C] Xishun Wang, Tong Su, Fang Da, Xiaodong Yang, “ProphNet: Efficient Agent-Centric Motion Forecasting with Anchor-Informed Proposals”, CVPR 2023

[D] Wenyuan Zeng, Ming Liang, Renjie Liao, Raquel Urtasun, “SYSTEMS AND METHODS FOR ACTOR MOTION FORECASTING WITHIN A SURROUNDING ENVIRONMENT OF AN AUTONOMOUS VEHICLE”, US Patent 2023/0347941

3. For cases 2. and 3. above, it may also be possible to fill the information gap by extrapolating previous camera detections based on either a simple linear model or, where possible, an object-class-specific realistic motion model.

While extrapolation can help to predict object motion it suffers from two fundamental limitations, both of which are addressed when using an event camera:

First, in the absence of additional information, extrapolation cannot generate a certifiable hypothesis of object motion, and will inevitably ignore (i) unmeasurable accelerations, jerks of objects, (ii) sudden appearances or disappearances of objects and (iii) sudden deformation of objects. This issue is especially grave in modeling pedestrians, which (i) are frequently subject to sudden, complex, and reflexive motion, (ii) can suddenly appear or disappear in the field of view, including their body parts like arms and legs, and (iii) have deformable appearances like when they stretch their arms, stumble or fall. An auxiliary event camera can fill this information gap, by gathering a continuous, low-latency stream of information.

We quantified the performance degradation of an image-based method based on this first limitation in Fig. 3 (a). There we show that predicting blindly into the future, without extrapolation degrades the performance of the image-based method from 43.5 at the start of the window to 32.7 mAP after 50 ms, a drop of 10.8 mAP. This degradation can be ameliorated by using an extrapolation technique (as suggested by the reviewer in

comment 4, and explained in more detail there) to 37.1 mAP, only a 6.4 mAP drop. By contrast using events completely eliminates this drop, and even leads to a 0.7 mAP performance increase to 44.2 mAP. The difference between the event-based and image-based+extrapolation methods at time $t=50$ ms shows the improvement on objects with non-linear motion, or deformation since at the end of the interval no linear interpolation inaccuracies arise. This shows that event cameras effectively address cases 2 and 3. We summarize these arguments in Sec. 3.3.

A second limitation is that, extrapolation requires fitting a motion model on a series of past detections, and this fit improves as more detections are used. However, a high number of reliable detections is rarely available for suddenly appearing objects like pedestrians, or only available after a significant time delay, due to framerate limitations (up to 500 ms in [A]). Event cameras generate a continuous stream of events, and therefore events can be used to generate detections at up to 5,000 Hz. As a result, an event-based detector will have gathered two orders of magnitude more detections compared to a 30 FPS sensor in a given time interval and can thus significantly improve extrapolation.

[A] Alexander Cui, Sergio Casas, Kelvin Wong, Simon Suo, Raquel Urtasun, “GoRela: Go Relative for Viewpoint-Invariant Motion Forecasting”, ICRA 2023.

4. The authors already use an interpolation strategy to generate the ground truth bounding boxes for image time stamps between two recorded frames. The authors may wish to consider using a similar approach directly to generate in-between data.

We thank the reviewer for this suggestion, which we implemented for our baseline that uses only images. As suggested, we use the detections in the past and current frames to fit a linear motion model on the object's position, height, and width. As our method is an object detector, we need to establish associations between past and current objects. We did this in the following way: For each object in the current frame, we selected the object of the same class in the previous frame with the highest IOU overlap and used it to fit a linear function on the bounding box parameters (height, width, x position, and y position). If no match was found (all IOUs were 0 for the selected object), it was not extrapolated, but instead kept constant. Using this technique we updated Fig. 3 (a), and the new baseline method is colored in brown.

In the inter-frame detection experiment, the results indicate that object extrapolation can indeed be used to enhance the prediction of RGB-based methods in the blind time between frames, yet it still loses 6.4 mAP in going from 43.5 mAP after the first frame to 37.1 mAP after 50 ms. For reference, the image baseline decreases from 43.5 to 32.7 mAP, a drop of 10.8 mAP, and our method *increases* in performance from 43.5 to 44.2 mAP. We report these results in Sec. 3.3 and describe the baseline in Sec. 5.5.3 “Image-based Methods”.

5. It may also be the case that extrapolation, which is not a new certified snapshot of reality, could lead to larger errors - in which case it would be more appropriate to use the event information. Which one to use in which situation would also be an interesting question.

Here we summarize the findings in the above answers to address this question. Event cameras provide needed additional information to provide certified snapshots in the following situations:

- a. Objects suddenly appear or disappear within one frame. This includes the sudden appearance or disappearance of object parts like legs, arms, or tires. This was validated experimentally in comment 1.
 - b. Objects undergo sudden changes in acceleration jerk, or morphology. This includes sudden complex and reflexive motion of pedestrians, like sudden freezing, stumbling, falling, or turning. This was validated experimentally in comment 2.
 - c. Finally, we argue that events from an event camera provide motion information that encodes intent and motion trajectories better than partial observations, or provide over two orders of magnitude more detections for extrapolation compared to standard image-based sensors.
6. The authors used YoloX as the base model for comparisons. As the newer models such as YoloV7 or YoloV8 outperform YoloX by at least 3% for mean average precision, the authors' comments on whether this would affect the comparisons would be highly appreciated.

Replacing the detector would improve almost all baselines, but would render the comparisons the same. We selected YOLOX for its beneficial speed compared to newer models.

7. Additional aspects which should be addressed in the opinion of the reviewer: Handling of stationary and relatively stationary objects

Our hybrid model combines events and frames and thus handles stationary objects better than a purely event-based method. This is because the detector can leverage the images, which can still capture objects, despite them not being visible in the events.

8. Statistics for edge cases:

For statistical analysis, we will define an edge case as an image that contains at least one appearing or disappearing object, which presumably would be missed by using a purely image-based algorithm. We found that this proportion is 31% of the training set and 30% of the test set. Moreover, we counted the number of objects that suddenly appear or disappear: We found that in the train set 4.2% of objects disappear, and 4.2%

appear, while in the test set, 3.5% appear, and 3.5% disappear. We report these figures in Sec. 5.5.1 “Statistics of Edge Cases”.

9. Time synchronisation requirements between camera frames and events

Events and frames were hardware synchronized via an external computer that sent trigger signals simultaneously to the image and event sensor. While the image sensor would capture an image with a fixed exposure upon triggering, the event camera would record a “special event” which exactly marked the time of triggering. We assign the timestamp of this event (and half an exposure time) to the image. We found that this synchronization accuracy was on the order of 78 microseconds, which we determined by measuring the mean squared deviation of the frame timestamps from a nominal 50,000 microseconds. We summarize these findings in Sec. 5.5.1, “Comments on Time Synchronization”.

10. Communication network implications and event transport delays

As discussed, we estimate the mean synchronization error on the order of 78 microseconds with hardware synchronization. Moreover, in a real-time system the event camera will experience event transport delays which are split into a “maximal sensor latency”, “MIPI to USB transfer latency” and a “USB to computer transfer latency”. For the Gen3 sensor the sum of all worst case latencies can be as low as 6 milliseconds. It can be further reduced by using directly a MIPI interface in which case this latency reduces to 4 ms. However, this worst-case delay is only achieved during static scenarios, where there is an exceptionally low event rates such that MIPI packets are not filled sufficiently. However this case is almost never achieved due to the presence of sensor noise, and also does not affect dynamic scenarios with high event rates. Finally, note that while all three latencies would affect a closed-loop system, our work is evaluated in an open loop and thus does not experience these latencies, or synchronization errors due to these latencies. We summarize this discussion in Sec. 5.5.1 “Comments on Network and Event transport Latencies”.

Comments by Referee #2

1. C. the authors use available datasets and created a dataset that combines event and rgb data. The ground truth (GT) boxes positions is approximated linearly between frames, which introduces small errors.

Note that errors due to linear interpolation only affect the interframe evaluation in Fig. 3 (a), and not overall results Tab. 2 which are evaluated after 50 ms when full ground truth bounding boxes are available. The interframe interpolation merely indicates that our

method can achieve stable performance in the blind-time between frames, and overcome the limitations of frame-based methods that drop by up to 10.8 mAP. To validate the accuracy of the ground truth, we regenerated it using cubic and linear interpolation. To perform cubic interpolation we needed to limit the test set to samples that do not have appearing or disappearing objects, and have tracks that are at least four frames long. On this subset, the object detection performance only fluctuated within a maximal deviation of of 0.2 mAP, indicating that the ground truth is accurate enough.

Nonetheless, results both from Tab. 2 and the end-points in Fig. 3 (a), i.e. performances at times 0 and 50 ms, support the finding that our method excels at performing non-linear interframe object detection, and were found with ground truth that does not suffer from linear interpolation artefacts. We summarize this discussion in Sec. 5.5.1 “Comment on Approximation Errors due to Linear Interpolation”, and visualize the findings in Fig. 11.

2. An another source of error is the alignment based on an homography between frame and event cameras. Nonetheless, the quality of the data seems sufficient for the task, that does not require a pixel-level precision as described. If the system is used for decision making like in an autonomous car more precise datasets might be required.

Throughout the dataset, event-to-image misalignment is small and never exceeds 6 pixels, and this is further supported by visual inspection of Fig. 15. Nonetheless, we characterize the accuracy that a hypothetical decision-making system would have if worst-case errors were considered. Consider a decision-making system that relies on accurate and low-latency positioning of actors like cars and pedestrians. Such a system could use the proposed object detector (using the small-baseline stereo setup with an event and image camera) as well as a state-of-the-art event camera-based stereo depth method [A] (using the wide-baseline stereo event camera setup) to map a conservative region around a proposed detection. Such a system would still have a low latency and provide a low depth uncertainty due to a low disparity error of 1.2-1.3 pixels,

characterized on DSEC in [A]. We can calculate the depth uncertainty $\sigma_D = \frac{D^2}{b_w f} \sigma_d$

With a disparity uncertainty $\sigma_d = 1.3$ pixels, the depth D at 3 meters, the focal length at $f = 581$ pixels and the event camera to event camera baseline at $b_w = 50$ cm. This results in a depth uncertainty of $\sigma_D = 4$ cm. Likewise, the lateral error is $\sigma_p = \frac{D}{f} \sigma_d$.

For lateral positioning we can assume a disparity error that is bounded by the

misalignment between events and frames, which is $\sigma_d < \frac{f b_s}{D}$ where $b_s = 4.5$ cm is the small baseline. Inserting this uncertainty the resulting lateral uncertainty is bounded by

$\sigma_p = \frac{D}{f} \sigma_d < \frac{D}{f} \frac{f b_s}{D} = b_s$, which means $\sigma_p < 4.5$ cm. These numbers are well within the tolerances of automotive systems which typically expect a 3% of distance to target

uncertainty which for 3 meters would be 9 centimeters. Moreover, This lies within the tolerance of current agent forecasting methods [B, C, D] that are currently finding their way into commercial patents [E], where we see displacement errors in prediction on the order of 0.6 meters, more than one order of magnitude higher than the worst-case error of our system.

Finally, we argue that despite the misalignment, our object detector learns to implicitly realign events to the image frame, due to the training setup. Since the network is trained with object detection labels that are aligned with the image frame, and slightly misaligned events, the network learns to *implicitly realign* the events to compensate for the misalignment. Since the misalignment is small, this is indeed simple to learn. To test this hypothesis, we used the LiDAR scans in DSEC to align the object detection labels with the event stream, i.e. in the frame it was not trained for, and observed a performance drop from 41.87 to 41.86 mAP. First, the slight performance drop indicates that we are moving the detection labels slightly out of distribution, thus confirming that the network learns to implicitly apply a correction alignment. Second, the small magnitude of the change highlights that the misalignment is small. We reproduce this discussion in Sec. 5.5.1 “Comment on Event-to-Image Alignment”.

[A] Hoonhee Cho, Jegyeong Cho, and Kuk-Jin Yoon, “Learning Adaptive Dense Event Stereo from the Image Domain”, CVPR 2023

[B] Alexander Cui, Sergio Casas, Kelvin Wong, Simon Suo, Raquel Urtasun, “GoRela: Go Relative for Viewpoint-Invariant Motion Forecasting”, ICRA 2023.

[C] Zikang Zhou, JianPing Wang, Yung-Hui Li, Yu-Kai Huang, “Query-Centric Trajectory Prediction”, CVPR 2023

[D] Xishun Wang, Tong Su, Fang Da, Xiaodong Yang, “ProphNet: Efficient Agent-Centric Motion Forecasting with Anchor-Informed Proposals”, CVPR 2023

[E] Wenyuan Zeng, Ming Liang, Renjie Liao, Raquel Urtasun, “SYSTEMS AND METHODS FOR ACTOR MOTION FORECASTING WITHIN A SURROUNDING ENVIRONMENT OF AN AUTONOMOUS VEHICLE”, US Patent 2023/0347941

3. D. The statistical uncertainty of the results is not described, even though is rarely provided in similar works.
4. F. I would define the perceptual and computational latency to avoid confusion.

In our work, we define perceptual latency as the time between impacting photon to data read-out, which for the Prophesee Gen4 event camera is roughly 0.2 milliseconds. By contrast, computational latency is the time taken to process the data to generate an output detection and is hardware-dependent. We place these definitions after the first uses of the terms in Sec. 1.

5. It would be interesting also to consider a real scenario, where the dense features are obtained only after the frame rgb acquisition and after dense computation. Therefore the dense features are outdated by a certain latency. It is not clear whether such realistic scenario has been considered and if there are implications in the results.

Following the suggestion of the reviewer, we considered such a scenario and implemented the following modification to our method: After a computation time Δt for computing the dense features, we start integrating events from the interval $[\Delta t, 50 \text{ ms}]$ in to the detector. This means that for time $0 < t < \Delta t$ no detection can be made, as no features are available from images. In this interval either the event-only method from Tab. 6 (first row) can be used, which does not use images, or a linear propagation from the detection from the previous interval. We evaluate the performance of our method with a ResNet-18, ResNet-34, ResNet-50 + detection head. These CNNs had computational latencies of 5.3 ms, 8.2 ms and 12.7 ms respectively on a Quadro RTX 4000 laptop GPU. We report the results in Tab. 6. We see that compared to our method without latency consideration, the method achieves 0.3 mAP lower performance after 50 ms when evaluated on the full test set. On the subset used for interframe detection evaluation, after 50 ms the performance degrades from 44.2 mAP to 43.8 mAP, still 6.7 mAP higher than 37.1 mAP which is the performance of the image-based baseline with extrapolation after 50 ms. On the full test set the degradation for ResNet-34 is 0.1 mAP, and for ResNet-18 is 0.1 mAP. Notably for a smaller model the degradation is smaller since the latency is also smaller. Nonetheless, the largest model still achieves the best results after 50 ms.

This demonstrates, that our method outperforms image-based ones even when considering computational latency due to CNN processing. Nonetheless, to minimize the effect of this latency future work could consider incorporating the latency into the training loop, in which case the method will likely learn to compensate for it. We report this experiment in Sec. 8 “Incorporating CNN latency into the prediction”.

6. It would also be interesting to have time performance and not just Mflops in the various tables. Obviously, the time is hardware-dependent, but it gives an idea, especially on how much parallelization is possible (MFlops do not give an idea on how many of such flops can be done in parallel).

We have added runtime performances of our method in Tab. 2, and find the fastest Method to be DAGr-S+ResNet50 with 9.6 ms. As stated by the reviewer, specific hardware implementations are likely to reduce this number substantially. Moreover, as can be seen in the comparison, MFLOPS/ev does not correlate with runtime at these low regimes, and this indicates that significant overhead is present. Indeed we are using the PyTorch Geometric library [A], which is optimized for batch processing and thus introduces data handling overhead. When eliminating this overhead, runtimes are expected to decrease even more.

[A] Fey, Lenssen, "Fast Graph Representation Learning with PyTorch Geometric", ICLR Workshop on Representation Learning on Graphs and Manifolds, 2019

7. typos and remarks: page 5: the "s" seems a typo: FLIR BlackFly s global shutter,

Indeed there was a typo. The name of the model is FLIR BlackFly S.

8. page 5: in this section is not entirely clear that the performance is compared (I assume) over time: "We also observe that all methods slightly increase, reach a maximum, and then decrease again, improving the initial score at $t = 0$ by between 0.6 and 0.7 mAP. The performance increase can be explained due to the addition of events". I suggest to make it more explicit.

page 11: double dots: high accuracy and efficiency..

The reviewer is indeed correct that this is compared over time. We have made it more explicit in the revision. We have edited Sec. 3.3 accordingly.

9. G. the references are appropriate and they properly credit previous work for both event and frame-based data. If possible I would add some reference to other point cloud methods, for example Flex-convolution and PointNet. They might be relevant for the related work.

PointNet++: Deep Hierarchical Feature Learning on Point Sets in a Metric Space Hao Su, Charles R. Qi, Li Yi, Leonidas J. Guibas

Flex-Convolution (Million-Scale Point-Cloud Learning Beyond Grid-Worlds) Fabian Groh, Patrick Wieschollek, Hendrik P.A. Lensch

We will gladly add these works, as they are highly relevant for point-cloud-based processing. We have added them to Sec. 6.

10. H. I would make the contributions more explicit in the introduction, e.g. the pruning. The rest of the text is overall clear.

We have emphasized the contributions more strongly, following the suggestion. We have made the edit in Sec. 1.

Reviewer Reports on the First Revision:

Referees' comments:

Referee #1 (Remarks to the Author):

The reviewer acknowledges and appreciates the efforts made by the authors to thoroughly address my feedback and to incorporate appropriate revisions into their manuscript.

In their revisions and responses, the authors have attempted to strengthen the rationale for using event-triggered cameras, especially in situations of partial observability - when image-based detectors must either wait for the next frame cycle or for full observability to be restored to update their state information. The reviewer acknowledges the additional experiments included by the authors that demonstrate the improvement in predictions when using a combined event and image-based detector in such cases. While an improvement in the Mean-Average-Precision (mAP) metric is to be expected with an additional (reliable) source of information in the system, the interpretation of this metric becomes critical in assessing the impact of the improved mAP. The reviewer would therefore ask the author to consider providing a qualitative interpretation of the mAP difference for the application.

The authors have also provided implementation details of the synchronisation mechanism between the event and frame sensors using a hardware trigger mechanism with an approximate synchronisation accuracy of 78 microseconds. As multi-sensor systems are increasingly using network-based time synchronisation such as IEEE1588v2 instead of hardware triggering due to concerns about scalability and ease of expansion, the effect of time synchronisation degradation on detection quality would be of interest to most researchers working in this area. The reviewer would like to ask the authors to consider adding this information to the time synchronisation chapter.

Referee #2 (Remarks to the Author):

A. the authors propose a method to fuse RGB images and events from event-cameras for object detection. The results shows the increase of mean average precision with respect to object detectors based on RGB or events singularly, in particular during interframe detections, while keeping the bandwidth and latency low.

For RGB they use resnet-like architectures and for events they use a asynchronous graph CNN called Deep Asynchronous Graph Neural Network (DAGr). They also propose strategies such as pruning to reduce computation on Graph CNN.

B. The paper is original in the sensor fusion aspect combining state of the art methods for RGB object detection and event-based object detection, and novel in the strategies to improve the performance of the DAGr.

C. the authors use available datasets and created a dataset that combines event and rgb data. The ground truth (GT) boxes positions is approximated linearly between frames, which introduce small

errors. However, in the section 3.3 the authors evaluate the ability of the method to measure both linear (between the interval) and non-linear motions (at $t = 50\text{ms}$). It is possible to evaluate non-linear motion only because at 50ms a detection is available in the GT, otherwise evaluate non-linear motion is not possible when the GT is estimated as a linear interpolation between frames.

Another source of error is the alignment based on an homography between frame and event cameras. The authors added a discussion in section 5.5.1 on the amount of error introduced which is relatively small.

The approach is sound and the presentation is good with several graphs.

D. The statistical uncertainty of the results is not described, even though is rarely provided in similar works.

E. The conclusions are correct highlighting the advantage of the proposed approach and also the limitation on standard hardware. The conclusions are based on several ablation studies therefore they seem robust and reliable.

F. As suggested during the review the authors made the definition of perceptual and computational latency, and added a real scenario, where the dense features are obtained only after the frame rgb acquisition and dense computation. The authors therefore discuss the degradation introduced by such latency.

As suggested during the review the authors added the time performance and not just Mflops.

typos and remarks:

-the description of "with and without extrapolation" used for the image-based baseline seems missing. I assume that when without extrapolation the detection is kept constant between the frame and the next. If so the terminology is confusing: "without extrapolation" is actually an extrapolation with a constant model, "With extrapolation" is an extrapolation with a linear model. I would explain better this part.

- there are 2 REFERENCES sections, is this a mistake?

- section 3.3: the definition of t_E and t_I in the text " $\Delta t = t_E - t_I = 50\text{ms}$ " seems missing

- there are references to a supplementary material, I do not see it in the paper.

- table 2: what is speed in tab. 2? (since the other statistics are per event inserted, the speed is not clear in which condition is computed. is it the average on the dataset?)

It is also not clear the connection between tab. 2 and tab. 6. Is the table 2 speed without dense features computation? is table 6 speed only the dense computation? this needs to be clarified.

G. the references are appropriate and they properly credit previous work for both event and frame-based data. As suggested the references to other point cloud methods were added.

H. text is overall clear; as requested the contributions are more explicit in the introduction.

Author Rebuttals to First Revision:

Comments by Referee #1

1. While an improvement in the Mean-Average-Precision (mAP) metric is to be expected with an additional (reliable) source of information in the system, the interpretation of this metric becomes critical in assessing the impact of the improved mAP. The reviewer would therefore ask the author to consider providing a qualitative interpretation of the mAP difference for the application.

The mean average precision measures a weighted average of precisions at each detection threshold. At each threshold, the weight corresponds to the increase in recall from the previous threshold. The mAP is thus maximized if a method retains a high precision, while the recall curve undergoes a steep increase. We observe that using an event camera mostly aids in increasing the recall slope. The recall slope is increased since the addition of an event camera improves object localization between frames. This improvement in localization entails a higher intersection over union and thus reduces false negatives at high thresholds. Reducing false negatives contributes to increased recall. We added this explanation to the end of Sec. 3.3.

2. As multi-sensor systems are increasingly using network-based time synchronisation such as IEEE1588v2 instead of hardware triggering due to concerns about scalability and ease of expansion, the effect of time synchronisation degradation on detection quality would be of interest to most researchers working in this area. The reviewer would like to ask the authors to consider adding this information to the time synchronisation chapter.

As suggested, we studied this dependency by offsetting the event timestamps by a fixed synchronization delay $\Delta t = -20, -15, \dots, 15, 20$ milliseconds. We then evaluated our largest model on the test set and report the results in Fig. 11. The results show that the mAP of our model only degrades by 0.2 mAP, and 0.5 mAP for -20 and 20 milliseconds latency respectively. This highlights, that our method is robust to small synchronization delays. We added this experiment to Sec. 5.5.1

“Comments on Network and Event transport Latencies”. This conclusion is further supported by the experiment in Sec. 8 *“Incorporating CNN latency into the prediction”*. There, we simulate a maximum delay of 12.7 milliseconds between events and image features. While before this delay played the role of the CNN processing latency, it can also play the role of network latency. We see that despite a high delay (max 12.7 ms) mAP is only weakly affected, degrading from 41.9 mAP to 41.6 mAP, a 0.3 mAP reduction.

Comments by Referee #2

1. the description of "with and without extrapolation" used for the image-based baseline seems missing. I assume that when without extrapolation the detection is kept constant between the frame and the next. If so the terminology is confusing: "without extrapolation" is actually an extrapolation with a constant model, "With extrapolation" is an extrapolation with a linear model. I would explain better this part.

Indeed, the reviewer is correct in making this distinction. We have clarified this point in Sec 5.5.3 *“Image-based Methods”*, and Sec 3.3. We also updated the legend and caption of Fig. 3 A.

2. there are 2 REFERENCES sections, is this a mistake?

As per Nature formatting guidelines, we created a separate reference section for the main paper and the method section.

3. section 3.3: the definition of t_E and t_I in the text " $\Delta t = t_E - t_I = 50\text{ms}$ " seems missing

We added this definition in Sec. 3.3.

4. there are references to a supplementary material, I do not see it in the paper.

We apologize for the confusion. We meant to reference the methods section.

5. table 2: what is speed in tab. 2? (since the other statistics are per event inserted, the speed is not clear in which condition is computed. is it the average on the dataset?)

It is indeed the average over the dataset. We have added clarifications in the captions of Tab. 2 and 6.

6. It is also not clear the connection between tab. 2 and tab. 6. Is the table 2 speed without dense features computation? is table 6 speed only the dense

computation? this needs to be clarified.

In Tab. 6 we only measure the time needed for dense computation by the CNN, while in Tab. 2 we measure the time needed to insert a single event into the GNN.